# Validation of a modeling methodology for wind turbine rotor blades based on a full scale blade test

Pablo Noever-Castelos[1], Bernd Haller[2], and Claudio Balzani[1]

[1]Leibniz University Hannover, Institute for Wind Energy Systems, Appelstr. 9A, Hanover, 30167, Germany
[2]Fraunhofer Institute for Wind Energy Systems IWES, Am Seedeich 45, 27572 Bremerhaven, Germany

**Correspondence:** Pablo Noever-Castelos (research@iwes.uni-hannover.de)

**Abstract.** Detailed 3D finite element simulations are state-of-the-art for structural analyses of wind turbine rotor blades. It is of utmost importance to validate the underlying modeling methodology in order to obtain reliable results. Validation of the global response can ideally be done by comparing simulations with full scale blade tests. However, there is a lack of test results for which also the finite element model with blade geometry and layup, as well as the test documentation and results are completely available.

The aim of this paper is to validate the presented fully parameterized blade modeling methodology that is implemented in an in-house model generator, and to provide respective test results for validation purpose to the public. This methodology includes parameter definition based on splines for all design and material parameters, which enables fast and easy parameter analysis. A hybrid 3D shell/solid element model is created including the respective boundary conditions. The problem is solved via a commercially available finite element code. A static full scale blade test is performed, which is used as the validation reference. All information, e.g., on sensor location, displacement and strains, are available to reproduce the tests. The tests comprise classical bending tests in flapwise and lead-lag directions according to IEC 61400-23 as well as torsion tests.

For the validation of the modeling methodology, global blade characteristics from measurements and simulation are compared. These include the overall mass and center of gravity location, as well as their distributions along the blade, bending deflections, strain levels, and natural frequencies and modes. Overall, the global results meet the defined validation thresholds during bending, though some improvements are required for very local analysis and especially the response in torsion. As a conclusion, the modeling strategy can be rated as validated, though necessary improvements are highlighted for future works.

## 1   Introduction

Rotor blades are major components of wind turbines. They are susceptible to damages, which, in case they need repair, can result in severe turbine downtime (Reder et al., 2016). It is thus crucial to develop a blade design that withstands all expected loads without damage. Though a blade prototype is always tested at the full blade scale in the certification process (International Eletrotechnical Comission, 2014), such tests are very costly and time-consuming, especially for growing blade dimensions (Ha et al., 2020). For this reason, full scale blade tests are executed one time only per blade design. Hence, a reliable and fast virtual blade design procedure is required. Full 3D finite element (FE) analysis is accurate, but computationally expensive. A widely

used approach for wind turbine blade design is to carry out two-dimensional cross-sectional analyses which offer a reduced level of complexity, but are a fast and efficient alternative for rotor blade pre-designs (Chen et al., 2010). Tools like VABS (Yu et al., 2002) or BECAS (Blasques and Stolpe, 2012) compute cross-sectional properties based on a 2D-FE-analysis, which are necessary to feed the aero-elastic models in order to recalculate the design loads on the turbine blades and close the design iteration loop. Nevertheless, at a final stage 3D FE analyses have to be performed in order to obtain a reliable blade design and account for structural details, such as adhesive joints, longitudinal geometric discontinuities, ply drops, or local buckling analysis, which are not considered in a 2D-FE-analysis.

## 1.1 State-of-the-art 3D Finite-Element Modeling of Wind Turbine Blades

Automated model creation is state-of-the-art and a key to enhancing the design process significantly by reducing computational time, increasing the possible number of design loops and avoiding modeling errors caused by the user during a manual model creation. Among a vast selection of common software tools originated from the scientific community, QBlade (Marten et al., 2013) for example focuses on the aerodynamic blade design, applying only an Euler-Bernoulli beam approach for the structure. Sandia's NuMAD (Jonathan C. Berg and Brian R. Resor, 2012) additionally contains a more sophisticated structural description taking into account a composite layup definition for the blades' sub-components. The same holds for the software package FOCUS developed by WMC Laboratories, now part of LM Wind Power (Duineveld, 2008), which is a state-of-the-art tool used for blade design in many engineering offices. In FOCUS the user discretizes stations in the span-wise direction with all necessary geometrical information of these particular cross-section and in between the tool interpolates linearly all missing data. Hence a high discretization of stations along the blade span (e.g., 45 stations for a 20 m blade) is necessary to correctly reproduce non-linear changing geometrical or material information in the span-wise direction.

Another more advanced tool is the optimization framework CP-Max, see Bottasso et al. (2014). The parameterization is based on mathematical functions for the blade design description in the span-wise direction. This method has the advantage of reducing the number of stations along the blade without losing information in between, while enabling the framework to efficiently manipulate the parameters during optimization. The focus of the optimization framework is to find the best compromise between accuracy and costs. A similar blade parameterization is used within the FUSED-Wind Framework (Zahle et al., 2020), which contains spline descriptions for each parameter as shown in the prominent example of the DTU 10MW reference blade design (C. Bak et al., 2013). An interface to the framework was later incorporated into the python-tool FEPROC and the correct modeling process was verified against the DTU 10MW reference blade (Rosemeier, 2018). Another blade modeling tool developed at Ghent University also relies on function-based descriptions of the blade parameters and focuses on a modular principle of Finite Element (FE) constellations for modeling the different blade components and joints in the structure (Peeters et al., 2018). The latter algorithm and CP-Max are able of generating solid element models, while the others rely on more common shell element representations.

A lot of scientific contributions deal with FE modeling, but focus on structural details such as trailing edge adhesive joints. Eder and Bitsche (2015) for instance use a local model with fracture analysis to deduce the debonding between shell and adhesive due to buckling and validate the behavior against experimental results. Ji and Han (2014) also apply fracture me-

chanics and use a detailed model at the shear web adhesive joint to analyze crack propagation in the bond line. Most of these locally detailed models are used within a global-local modeling approach like in Chen et al. (2014) to reduce the global model complexity while keeping a high level of detail at local spots.

## 1.2 Objectives of this paper

Though some of these model creation frameworks may work with functions or splines describing the blade's geometrical or layup information, most of them work with a reasonably high number of airfoils/stations that in addition to the blade's geometry yield the outer blade shape by a global linear or higher order interpolation between the airfoils.

The presented method combines and extends several aspects of the different aforementioned software packages. The benefits are:

- it generates airfoils independent from any neighboring geometry and uses the relative thickness distribution to position these along the span. This assures the geometry distribution, as it avoids any overshoot due to spanwise geometry interpolation.

- any parameter which may vary over the radius can be defined as spline, e.g., relative blade thickness, layer thickness, material density or stiffness.

- it enables flexible and easy parameter studies due to the simple parameter variation based on splines.

- it is designed for research, as different modules can be easily replaced by an alternative code, e.g., airfoil interpolation, adhesive modeling.

- it generates an FE-Model in MATLAB and provides already an interface to ANSYS APDL (ANSYS Inc., 2021) and BECAS (Blasques and Stolpe, 2012), however interfaces to other FE-software can easily be implemented.

Different FE modeling procedures can result in different deformation and stress solutions, though based on the same model parameters, see (Lekou et al., 2015). Hence, it is important to validate modeling strategies by comparing simulations with full blade tests, which is the aim of this paper. A quasi-static full scale blade test is performed, including not only bending tests in flapwise and lead-lag direction – as are usually executed in the context of blade certification (International Eletrotechnical Comission, 2014) – but also torsion tests. This allows for an exceptionally detailed and thorough validation. Unlike other blade tests reported in literature (Chen et al., 2021), (Chen et al., 2017), (Jensen et al., 2006), (Overgaard and Lund, 2010), (Overgaard et al., 2010), the aim of the tests in this work is not the validation of failure models. Hence, the blade is not loaded up to failure.

The paper focuses on measuring and validating primarily the global elastic response of the blade, expressed in terms of deflections, mass distribution, and modal characteristics. That is why no global-local modeling approaches nor very local mesh refinements are considered for this study. However, secondly the local behavior in terms of strain levels along defined cross-sections is measured and compared to find a qualitative agreement with the model. The blade under investigation is the

SmartBlades 2 DemoBlade, a 20 m blade from the SmartBlades 2 project (SmartBlades2, 2016-2020), which includes pre-bending and a pre-sweep towards the trailing edge. The blade is modeled with our in-house blade model creation tool MoCA (Model Creation and Analysis Tool for Wind Turbine Rotor Blades), taking into account some major manufacturing-related deviations. The test setup and the load introduction are approximated via a combination of suitable boundary conditions and multiple point constraints. The simulation results are thoroughly compared with the test measurements.

## 1.3 Outline

The modeling strategy is addressed in Sec. 2 and Sec. 3. The test setup is described in Sec. 4. The blade was cut into segments after the tests in order to accurately measure the mass distribution and the locations of the centers of gravity along the blade. These measurements are also described in Sec. 4. The simulation versus test comparison is reported in Sec. 5, followed by the conclusions in Sec. 6.

## 2  Model Creation Framework

A framework to automatically generate fully parameterized 3D FE models of wind turbine rotor blades from a set of parameters was developed at the Institute for Wind Energy Systems at Leibniz University Hannover. The purpose of this tool called MoCA (**Mo**del **C**reation and **A**nalysis Tool for Wind Turbine Rotor Blades) is to enable users to investigate and analyze different blade designs or design parameter variations in an easy way, including structural details such as, e. g., adhesive joints. The following Section presents a brief description of the framework.

MoCA is based on a set of input parameters categorized in *Geometry*, *Plybook*, *Structure*, and *Material*. In general all parameters that describe a distribution along the blade are stored as splines over the blade's arc length, but even material parameters may be varied over the blade arc if necessary by using a spline. The parameter set *Geometry* contains all information on the outer geometry of the blade, i. e., all involved airfoils and their positions along the blade as well as the distributions of the relative thickness, chord length, twist angle, threading point location, prebend and presweep. The *Structure* set is associated with the structural description of the blade. This includes the specification of shear webs, adhesive joints and additional masses as well as cross-sectional division points that are mainly used to subdivide cross-sections into different regions of interest. The *Plybook* parameters contain the stacking information of different composite layups used in the blade. The parameter set *Material* comprises all material properties assigned for the different materials. These can either be isotropic or anisotropic on the macroscopic scale. The user can also specify a composite material based on microscopic characteristics of the fiber and matrix constituents, which are then transformed to a laminate via the well-known rule of mixtures.

In the following Fig. 1-4, each block is labeled with an index, which will be used in the following description for reference to the blocks of the respective figures. The flowchart in Fig. 1 depicts the structure of the finite element creation procedure implemented in MoCA on the basis of the parameter sets described above. First, the blade segmentation, i. e., the discretization in the span-wise direction, is defined. For each blade segment edge, a cross-section of the blade is calculated (5-7) by evaluating the *Planform* data (1). Then a finite element discretization of the cross-sections (8) is executed using the information of the

*Structure* (2), *Material* (3), and *Plybook* (4) parameter blocks. At this stage, an interface to the BECAS (Blasques and Stolpe, 2012) software (9) can be utilized to calculate the full $6 \times 6$ stiffness and mass matrices of a beam model. However, since our aim is to create a 3D blade model (12), we continue with the finite element discretization in the span-wise direction (10-11) utilizing a hybrid shell element/solid element strategy. Therein, we use shell elements to model the composite laminates and solid elements for the adhesives. The 3D FE mesh includes the node-to-element connectivity and elemental material assignments. The boundary conditions are added and the FE model is translated to an input file for the finite element solver of choice (13), which in our case is ANSYS Mechanical (ANSYS Inc., 2021). In the following, we describe the different steps of this overall procedure in more detail.

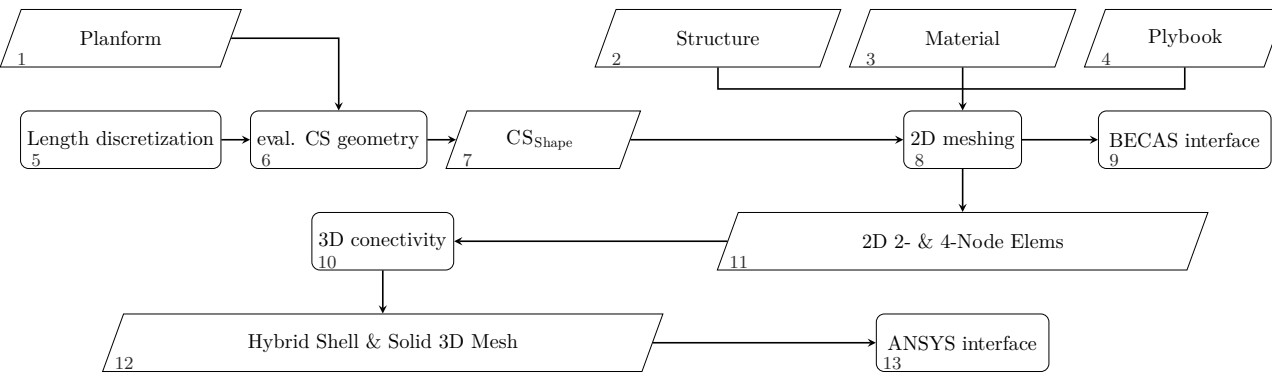

**Figure 1.** Flowchart of the finite element model creation procedure in MoCA.

Figure 2 visualizes the process of cross-section geometry calculation. After the blade segmentation, the *Geometry* data splines (2-7) are evaluated (8) for the particular blade arc positions (1) of the segment edges. According to the spline-based interpolation of the relative thickness $t_{rel}$ (10), an airfoil $AF$ is linearly interpolated (16) between the basic input airfoils (9), which have the next higher and lower relative thickness. In contrast to a global blade shape interpolation, the use of a blade independent airfoil interpolation enables the user to implement an own sub-function and replacing the former. The interpolated airfoils are then scaled by the chord length $c^*$ calculated via the respective spline (11,17), shifted along the chord to the correct threading point by the coordinate $tp^*$ (12,18), and twisted by the twist angle $\theta^*$ (13,19).

Until here, all transformations are performed in a 2D chord coordinate system with its final origin in the threading point. The cross-sections are now shifted to the correct 3D position (20), locating the 2D cross-sectional threading center on the prebended (14) and preswept (15) global blade axis. By doing so, the 2D chord coordinate system is still parallel to the blade root plane. Hence, the cross-sections are rotated by the slope angles of the prebend and presweep spline functions so that they are perpendicular to the threading axis. These shifted and rotated cross-sections are the final cross-sectional shapes denoted by $CS_{Shape}$ (21).

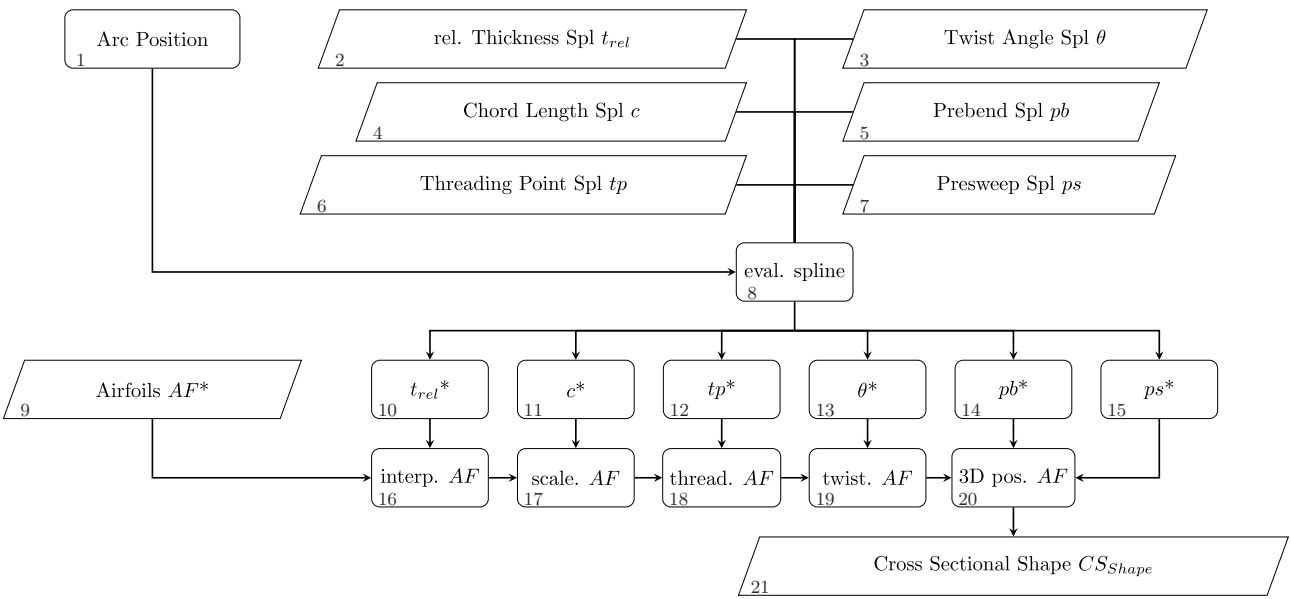

**Figure 2.** Flowchart of the calculation of the cross-sectional shapes $CS_{Shape}$.

According to Fig. 1, the next step is the 2D cross-sectional meshing, which is executed using the cross-sectional shapes $CS_{Shape}$ (7) and the parameter sets *Structure*, *Material*, and *Plybook* (2-4). This process is presented in Fig. 3. As before, all data (2-6) is evaluated (7) for the particular arc positions (1) at the blade segment edges. The division points are generated on the cross-sectional shapes (9,14). They serve to subdivide the cross-sections into regions of different material layups and are also used to define the positions of the shear webs. Then the shapes of the shear web (10,15) and the web and trailing edge adhesive joints (11,16) are computed. The computation of the blade's outer geometry and its structural topology is now finished. After inclusion of the *Material* (12) and *Plybook* (13) information, the FE discretization (18) on 2D cross-section level can be conducted. This yields either a two-dimensional mesh with 4-noded plane elements for the BECAS (DTU Wind Energy) interface (19) or a cross-sectional node map representing a hybrid 2D mesh with 2-noded elements for the composite laminates and 4-noded elements for the adhesives (20,21).

The last step in the creation of a 3D finite element model is to connect the 2D cross-sectional models, see Fig. 1 (10). The 2D line elements on the cross-sectional level yield 4-noded shell elements on 3D level after the 3D extension, and the 4-noded plane elements on cross-sectional level become 3D solid elements, respectively.

An additional module called *TestRig* is included in MoCA to model the boundary conditions similar to a full scale blade test. Full clamping of the blade root represents the geometrical boundary conditions, i. e., all degrees of freedom are fixed at the blade root. Figure 4 shows the process of the *TestRig* module for the introduction of force-like boundary conditions. In the real blade test, a number of load frames introduces loads that approximate the target bending moment distribution (or torsional

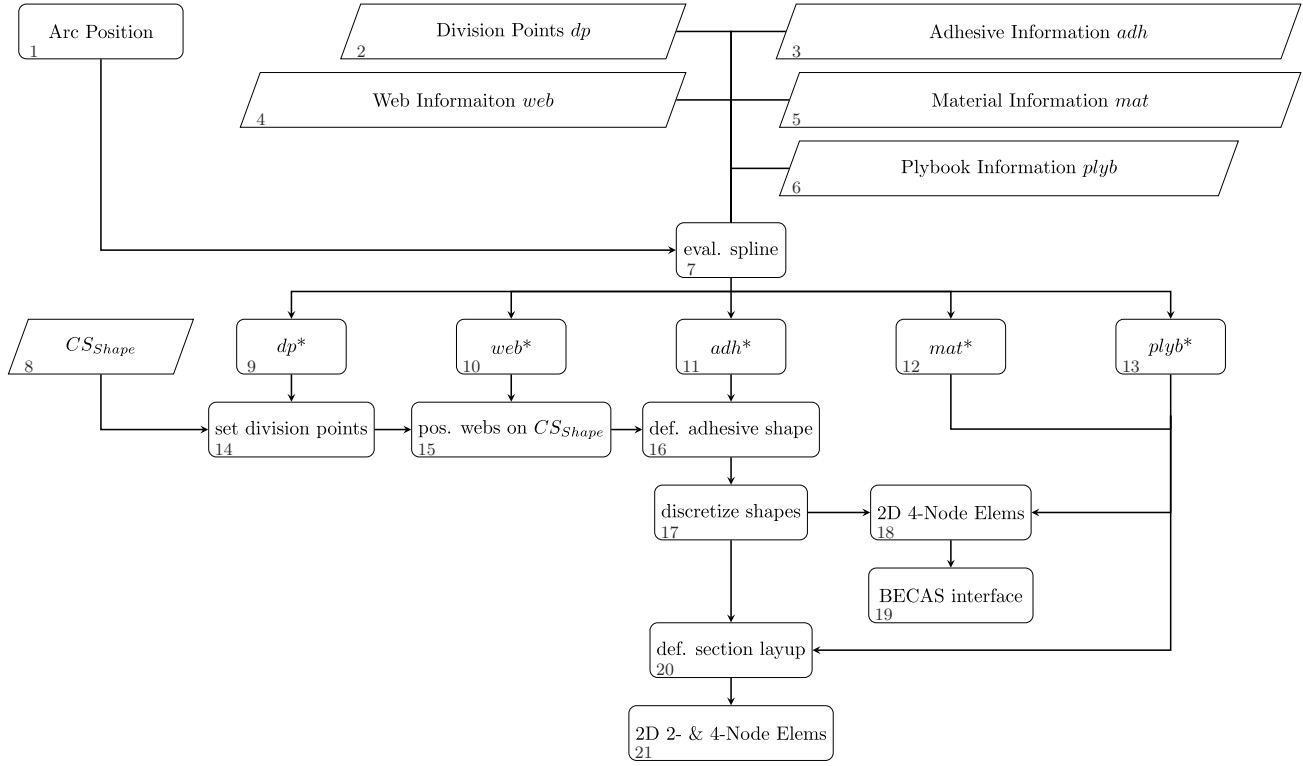

**Figure 3.** Flowchart of the 2D cross-sectional meshing routine in MoCA.

moment distribution, respectively). The *TestRig* module approximates the load frames by means of appropriate Multiple Point Constraints (MPC) and additional masses. For each load frame, the position along the blade (arc position) (1), the load frame width (4), the Center of Gravity (CoG) and the resulting mass (5) are specified as well as the load (2) and sensor points (3).

In the range where the load frame is located, MoCA searches all elements of the blade shell (6,9) and defines 2D slave elements (12) that share their nodes. An additional cross-section is created at the desired load frame position (7,10) according to the procedure depicted in Fig. 2. In this additional cross-section, the position of the load introduction (load point), the sensor points, and the center of gravity of the load frame are given in the blade coordinate system (11). These points are defined as master nodes (14). MPCs are included that connect the degrees of freedom of the master nodes and the slave nodes (13) by means of a rigid connection, i. e., there are no relative displacements between the master and the slave nodes. The additional mass of the load frame is applied to the CoG node (17), while the load is applied to the position where the load is introduced (16) in the real test (load point). In this way, we model solid and quasi-rigid load frames (15) and their effects on the blade response without adding detailed models of the load frames themselves, which is beneficial in the context of computational costs. The rigid connection implies that the deformation of the blade at the load frames is restricted. Similar to the real blade test according to IEC-64100 International Eletrotechnical Comission (2014), the load frames neighboring blade sections cannot

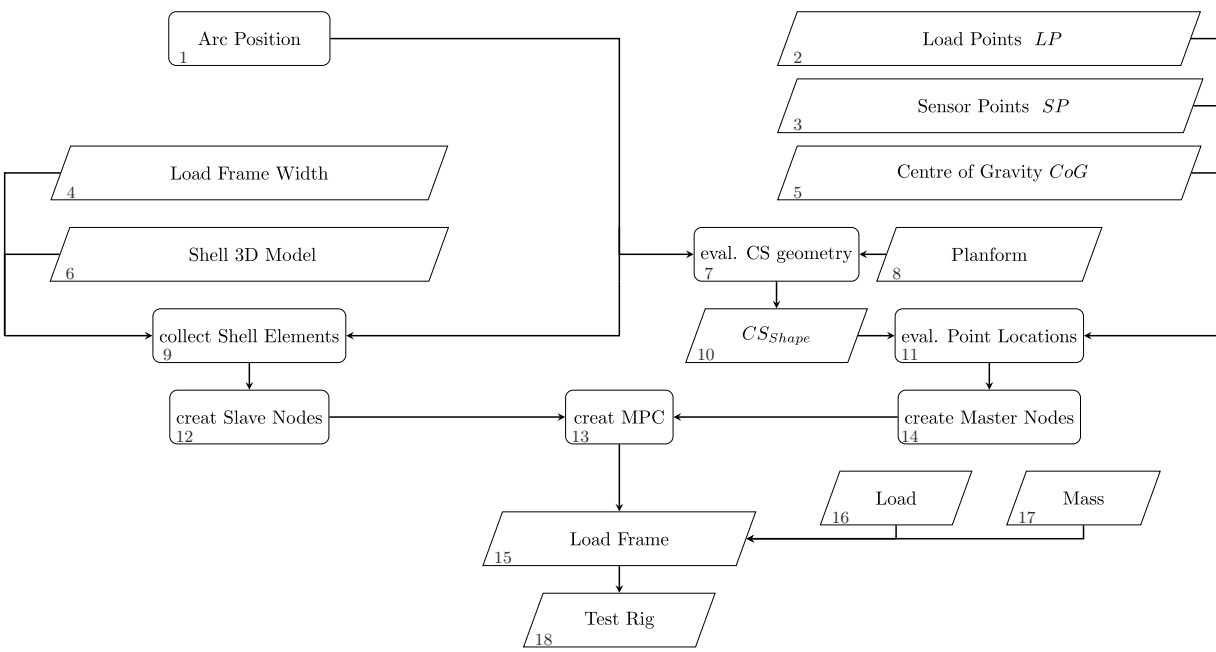

**Figure 4.** Flowchart of the procedure to model the boundary conditions in the *TestRig* module.

be evaluated, as the structural response is influenced by the quasi-rigid constraints.

The 3D finite element model including the mesh and the boundary conditions is translated to an input file for the finite element solver of choice via an integrated interface.

## 3 Modeling of the Test Blade

This Section briefly describes the blade under consideration, which is the *SmartBlades-DemoBlade*, a 20m long blade with prebend and presweep. It was designed and manufactured in the coordinated research projects *Smart Blades* (Teßmer et al., 2016) and *SmartBlades2* (SmartBlades2, 2016-2020). The blade is abbreviated by *DemoBlade* in the following.

The DemoBlade was designed to investigate bend-twist coupling effects in wind turbine rotor blades. Therefore a presweep of 1 m towards the trailing edge at the tip is intended to introduce a torsional twist into the blade. The offset between the aerodynamic centers of the swept airfoils and the pitch axis introduces a torsional moment and thus a torsional deformation, i. e., a twist in the outer part of the blade. The twist reduces the angle of attack of the respective airfoils and hence the aerodynamic coefficients. In this way the aerodynamic loads can be reduced.

During the manufacturing procedure and the latter testing, several different properties of the blade are captured. From these, the FE model only covers the geometrical deviations such as the chord, thickness and adhesive geometry deviations. However,

mass and stiffness adaption, to meet the measured natural frequencies and masses, or displacements will not be covered, as this would demand a thorough model updating procedure, which would go beyond the scope of this work. Therefore, this Section will refer to the geometrical measurements and the rest will be covered in Section 5


The full blade design of the DemoBlade *as designed* and the manufacturing documentation is available to the authors. In order to allow precise modeling of the DemoBlade *as built*, laser scanning of the blade mold was carried out to determine the geometry deviations. The derived chord length and absolute thickness distributions for the DemoBlade *as designed* and *as built* can be found in Noever-Castelos et al. (2021). Though the manufacturing deviations in the outer geometry are negligibly small

(Chord length $< 10$ mm and thickness $< 8$ mm), they will be considered in the modeling process.

After the full scale blade tests, the DemoBlade was cut into segments. The masses and the centers of gravity were determined for all blade segments. The respective procedure will be addressed later in this paper, see Sec. 4.4 and Sec. 5.5. Besides the weighing, the geometry was measured thoroughly in each cut cross-section in order to guarantee the correct positioning of the

shear webs in the FE model and to determine deviations from the design due to manufacturing errors. Especially the dimensions of the shear web/spar cap adhesive joints on the pressure side of the blade showed significant deviations to the blade design and had to be adjusted in the FE model. Figure 5, for instance, shows the cut at a radial position of 5.2 m ($r_{\text{norm}} = 26\%$). On the suction side (bottom) we see a nice, very thin, and over-laminated shear web/spar cap bonding. However, on the pressure side (top) the shear web/spar cap bonding (which was the blind bond, marked in red) is much thicker than specified in the design.

Moreover, there is a lack of adhesive in large portions of the blade, so that the shear web flanges were not covered entirely by adhesive material. This was actually found throughout the whole blade, where the thickness varied between 20 and 30 mm. The design defined a thickness of $9\pm3$ mm. Noever-Castelos et al. (2021) contains all the measured dimensions of the pressure side web adhesive.

In the FE model, we apply concentrated and line-distributed additional masses to cover any type of add-ons installed on the

blade such as the lightning protection cable or reflectors of an optical sensor system. Noever-Castelos et al. (2021) includes a Table with all additional masses and the respective modeling methods. Furthermore, MoCA predefines node positions in the blade that correspond to strain gauges installed on the blade. These are documented in Haller and Noever-Castelos (2021). They allow for accurate and easy extraction of strain results at the correct positions.

A mesh convergence study was performed in advance to ensure a satisfying mesh density. As stated before the purpose is to

validate primarily the global blade behavior and only secondary the local response. Therefore, no local mesh refinement will be performed, but the overall mesh density should yield acceptable convergence even at local level. Taking this into account, the convergence was first based on the global blade response in terms of total mass, center of gravity, tip deflection and the first 10 natural frequencies. Secondary the nodal strain results are examined for convergence at several positions covering the whole blade. The element dimensions are halved each step. At the finally chosen mesh size the deviations to the next step

are: global responses $< 1.5\%$, strains $< 2.1$ $\mu$-strains. It has to be stated, that for exact local strain measurements a modeling approach with solid-shells or layered solid elements is required to replicate the correct and detailed geometry of the structure.

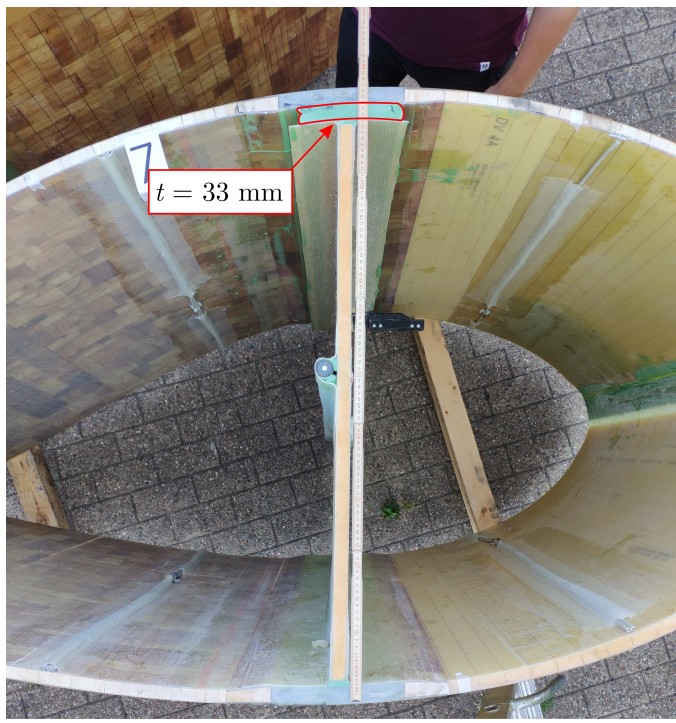

**Figure 5.** Cut cross-section at a radial position of 5.2 m ($r_{26\%}$) with a erroneous manufactured shear web/spar cap adhesive joint on the pressure side of the blade. The width should cover the complete web flange and the designed thickness is $9 \pm 3$ mm, however, the real thickness is measured to be 33 mm.

The resulting base model of the DemoBlade consists of 77,693 elements and 71,781 nodes. A total of 71,016 4-noded shell elements (SHELL181 elements in ANSYS), with offset nodes on the outer blade surface, represent the composite components and 6,260 8-noded solid elements (SOLID185 elements in ANSYS) model the adhesive joints. Fig. 6 depicts a cross-sectional view of the FE-Model at $r = 8$ m ($r_{norm} = 40\%$). All other elements are used to model additional masses in the blade. The only boundary conditions of the base model are the geometric boundary conditions at the blade root (full clamping as described above).

## 4   Test Description and Virtual Modeling

Several test configurations of the full scale blade test were performed to characterize the blade behavior under different load conditions and to prove that the blade design meets all requirements of the certification guidelines (International Eletrotechnical Comission, 2014). These configurations are than replicated in the virtual test setup and are described in this Section.

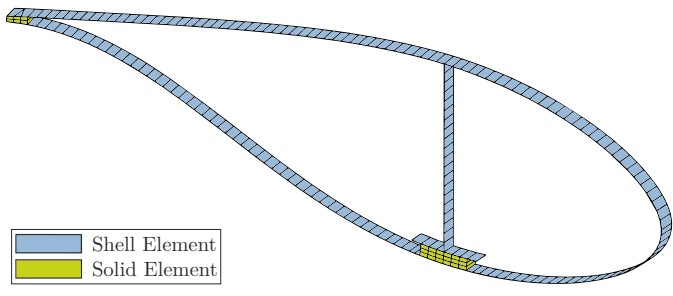

**Figure 6.** Cross-sectional cut of the FE-model at $r = 8$ m ($r_{\text{norm}} = 40\%$)

## 4.1 Mass and Center of Gravity

The first structural characterization considers the blade's mass and center of gravity (CoG). An indoor crane equipped with load cells at every hook lifted two points on each root and tip transport structure as shown in Fig. 7. As the blade remained still and
horizontally suspended the force at each suspension point and its radial position were recorded. After weighing individually the transport structures, the loading chains and the shackles, the weight was subtracted from the total recorded load at the measurement devices to obtain the total blade mass. Additionally, the mass of the blade bolts was subtracted from the total mass.

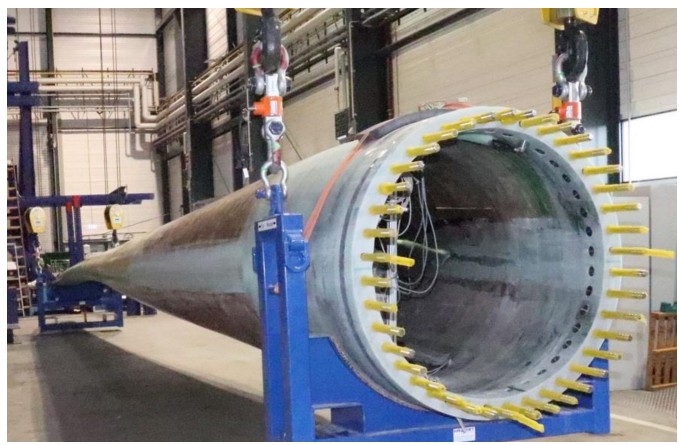

**Figure 7.** Setup for mass and center of gravity measurements.

The CoG is obtained by calculating the moment equilibrium with the measured loads with respect to a pivot point, in this
case the blade root center. This procedure was performed for the $z$-direction (along the span) and $y$-direction (along the chord).

The mass and CoG of the FE model is calculated during every analysis by default and can be extracted directly from the ANSYS log-file.

## 4.2 Modal Analysis

The experimental modal characterization was carried out by the German Aerospace Center (DLR) for different boundary conditions. The methodology is described briefly in the following. For details please refer to Gundlach and Govers (2019).

Free-free boundary conditions were applied after the blade manufacturing by means of elastic suspensions connected to lifting straps. The blade was excited using an impact hammer with soft tip at a total of 8 excitation points. Sensors distributed along the blade recorded the deformations, and the mode frequencies and shapes were extracted from the measurements.

The blade was then transported to Fraunhofer IWES and mounted on the test rig. The aim was a second modal characterization with the boundary conditions of the full scale blade test. Electrodynamic long stroke shakers were employed for the excitation of the blade, and sensor outputs were evaluated for the calculation of the mode frequencies and shapes.

During the FE modal analysis, the boundary conditions are adapted to the different characterization tests. In the free-free configuration, no boundary conditions are applied at all, partially resulting in zero eigenvalues related to rigid body motions. These are not considered in the validation process. For the test rig configuration, the blade root is fully clamped, i. e., all 6 degrees of freedom of the shell elements are fixed, for the sake of simplicity. Note that we neglect flexibility of the bolts and the test rig in this way, which we have to keep in mind when evaluating the simulation results.

## 4.3 Static Bending and Torsion Test Configuration

The SmartBlades2 DemoBlade was loaded with extreme loads in 4 directions. These four load cases correspond to maximum and minimum edgewise loading (MXMAX and MXMIN) as well as maximum and minimum flapwise loading (MYMAX and MYMIN). Furthermore, three static torsion tests were conducted, in which a torsional moment was applied only at one load frame at a time. The tests are referred to as MZLF2, MZLF3 and MZLF4, where LFX indicates the particular load frame, in which the torsional moment was introduced. The static tests provide the necessary information on the structural blade behavior required to validate the virtual model and test setup.

The tests were performed in the facilities of Fraunhofer IWES. The experimental quasi-static loading of the blade is accomplished with a series of horizontally mounted hydraulic cylinders. These are connected to the load cells via cables which are attached to the load frames mounted on the rotor blade. Each cable runs through pulleys that are mounted on the floor and redirect the forces from a horizontal to a vertical orientation. By attaching the load cells to the load frames (load point), the actual load applied to the rotor blade is measured and friction as well as weight of the loading cables do not affect the measurements. The general test setup is shown in Fig. 8.

In the following, some general information is given that is valid for all test setups. The test block angle (cone angle) is 7.5° upwards. The coordinate system referred to in this paper has its origin in the center of the blade root. The $y$-axis is facing vertical upwards, the $z$-axis points horizontally from the origin towards the blade tip (parallel to the floor, not to the pitch axis), and the $x$-axis follows from the right-hand rule (pointing left watching towards the tip). After turning the blade to the correct position and waiting for a static state, the signals of the load cells and the strain gauges are reset to zero. In the virtual test this is achieved by activating gravity, extracting the deformed nodal coordinates and taking these as the undeformed and stress-free

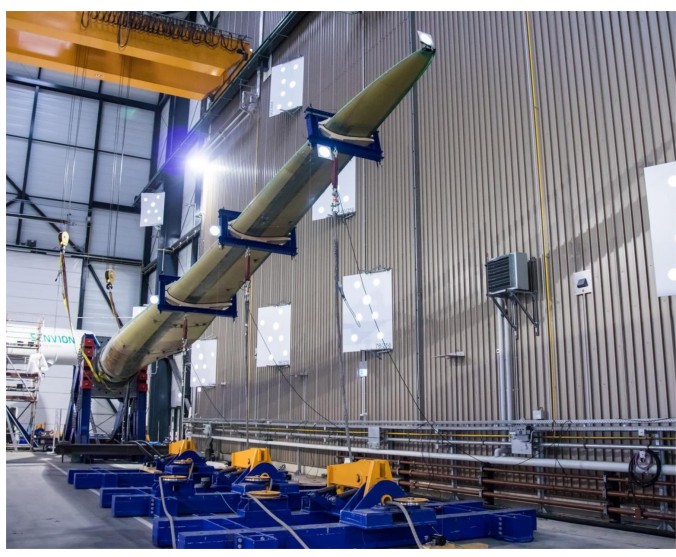

**Figure 8.** Photo of a static blade test configuration in flapwise direction.

state for the load tests. Gravity is thus not applied in the further analysis and the nodal displacements are virtually reset to zero so that it is easier to post-process the results. Preliminary verification showed that the corresponding error in the displacement is less than 0.5% with respect to a simulation that accounts for gravitation throughout the whole simulation.

In the tests, four steel load frames with wooden inlays that follow the blade shape at the respective span-wise positions are used to introduce the loads, see Haller and Noever-Castelos (2021). In the following, we refer to the load frames (LF) as LF1 (@ $r$ = 6.7 m, $r_{norm}$ = 33.5%), LF2 (@ $r$ = 9.7 m, $r_{norm}$ = 48.5%), LF3 (@ $r$ = 14.0 m, $r_{norm}$ = 70.0%), and LF4 (@ $r$ = 17.7 m, $r_{norm}$ = 88.5%), where $r$ denotes the span-wise position along the blade. Depending on the test setup, not all load frames are installed. Please refer to Noever-Castelos et al. (2021) to find an overview of all test setups. Each load frame is equipped with two eye-bolts to attach the load cables. These bolts are roughly positioned at the shear center position in the blade's cross-section to avoid unintended torsional loads. Detailed information on the load frames, such as mass, center of gravity, and the corresponding shear center position in the blade's cross-section are given in Haller and Noever-Castelos (2021).

The test setup is equipped with two different kinds of displacement measurements, an optical displacement measurement system and draw-wire-sensors (DWS). For the model validation in this paper, the DWS signals are considered. Using LINK11-Elements in ANSYS provides a simple and exact model of the draw-wires by defining the attachment points only. The deformation measured by the DWS is then modeled by the element-length variations of the link elements.

All necessary sensor positions (SP) and load introduction points (LP) on the load frames for the different test setups can be found in Haller and Noever-Castelos (2021). At each load frame position, either with or without installed load frame, two DWS are attached. One is connected to a point most to the front bottom corner, i. e., negative $y$-direction and one at the rear bottom corner, i. e., positive $y$-direction, of the load frames, or blade shells in case no load frame is installed. These two DWS

will be referred to as front and rear DWS in the following. At the blade tip, one DWS is attached referred to as Tip DWS. Note that during several load cases, one or the other load frame is not applied due to the setup design, thus the respective DWS have to be attached directly to the blade shell.

The angle between the loading cable and the blade axis can be adjusted in the experiment by changing the pulley block location within a discrete set of fixing points on the floor. Prior to the test setup, the optimal position for each pulley was determined based on the predicted blade deformation and the desired loading cable angle. The applied loads should be aligned to the load frame planes in the most deformed configuration. The DWS floor attachment and pulley block positions are specified for each test setup individually.

Additional to the DWS and the optical measurement system, several cross-sections along the blade are equipped with strain gauges, see Haller and Noever-Castelos (2021). The cross-sections at $r = 5$ m ($r_{norm} = 25\%$) and $r = 8$ m ($r_{norm} = 40\%$) are instrumented with strain gauge rosettes (bi-axial strain gauges) with 0°/90° and ±45° orientations. Fig. B1 and Fig. B3 depict the distributions, respectively. The angles 0° and 90° denote the span-wise and the cross-section-wise direction, wheres ±45° is defined accordingly. The 0°/90° rosettes are positioned every approx. 250-300 mm along the shell circumference. The ±45° rosettes are located at each web position as well as the leading and trailing edges. Details on strain gauge positions can be found in Haller and Noever-Castelos (2021).

All load cases have the same basic experimental procedure. They were designed to ensure that the actual test matches the specification requirements as closely as possible. Prior to each load case, the rotor blade is rotated to the desired position and mounted to the test stand (with the aforementioned 7.5° cone angle). The load cable pulley blocks are fixed to the appropriate fixation points on the floor. The load cells are installed between the load frames and the loading cables and are then connected to the data acquisition system. Each of the DWS is attached to the blade. The DWS base is positioned so that the wires run perpendicular to the floor. Finally, the loading cables are connected to the hydraulic cylinders.

The tests are then executed in the following order:

1. Functionality check of load cells and displacement sensors.
2. Compensation of load cell and strain gauge measurements (reset to zero).
3. Start data acquisition.
4. Ramp up loads until 100% of the target load, pausing at 40%, 60% and 80% partial loads for 10s each.
5. Ramp down loads, pausing at same load fractions as during ramp up.
6. Stop data acquisition and save measurement data to log file.

The process is similar in the simulation. Starting from the base model, which does not have a cone angle and the blade is positioned with the trailing edge pointing upwards, the steps are as follows:

1. Install necessary load frames.
2. Rotate blade around $z$-axis to desired position.
3. Include cone angle of test rig (incline the blade by 7.5° upwards around x-axis).
4. Apply gravity and extract new nodal coordinates.

335     5. Replace old nodal coordinates by the extracted new nodal coordinates (equal to resetting sensors to zero).

    6. Apply and ramp up loads onto the LINK11 elements acting as loading cables.

    7. Extract element length variation of the LINK11 elements acting as DWS for 40%, 60%, 80% and 100% of the target load.

All individual setups for the simulation with modifications to the base model, all necessary load frames, load points, sensor

340 positions, and forces as well as the corresponding ground positions of the pulley blocks and the DWS attachments are summarized in Haller and Noever-Castelos (2021). The ground position coordinates are given in the blade coordinate system of the base model (no cone angle, or rotation) described above at the beginning of this subsection.

In other torsion tests, e.g., Tiedemann and Chen (2021), a load is applied on a lever to introduce a combined torsion and

345 flapwise bending. With a subsequent test loading with pure flapwise bending only. The torsional deformation can then be found by subtracting the flapwise motion from the combined motion. However, our test setup follows the idea of introducing pure torsion by suspending the load frame approximately at the shear center location of the blade cross-section and inducing torsion by an offset load as shown in Fig. 9.

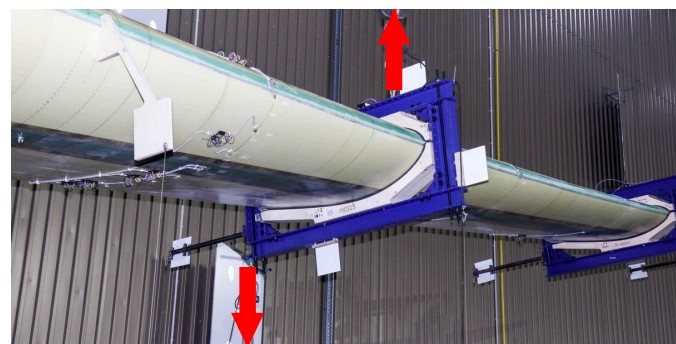

**Figure 9.** Configuration example of a static torsional loading on the blade with marked up and downwards facing forces.

Because the blade is still mounted at a block angle of $7.5°$ the axis of the torsional moment is not fully aligned with the pitch

350 axis, as the forces do not act exactly in the cross-sectional plane. The load cable oriented upwards was attached to a ceiling crane and to the load frame at approximately the shear center position. As the ceiling crane location is hard to record, but the load rope is perpendicular to the ground it was assumed that the location is 18m above ground ($y$-direction, approximately crane height). By this, deflections parallel to the floor, due to load application, would only result in small angle deviation of the perpendicular force. The force facing downwards was applied onto the load frame corner to create the lever with respect to the

355 shear center. Our procedure is similar to a combination of the pure torsion and locked torsion test presented by Berring et al. (2007). However, this method may imply some errors from:

    1. numerical shear center calculation

2. not suspending exactly at the cross-sectional shear center, but on the frame, which leads to an offset of the suspending force when the blade is twisted and thus an induced counteracting torsion.

3. no exact perpendicular downwards facing force

4. inclination of the blade

Regarding point 2, the offset of the load application point of the suspending cable from the numerically calculated shear center after twisting the blade, yield to 0.9%, 3.0% and 5.3% of the respective lever for the downwards facing force on LF2, LF3 and LF4, respectively. Theoretically, expecting a similar force pulling upwards as downwards, the induced torsion is reduced by the same relative values for the respective load cases MZLF2, MZLF3 and MZLF4.

The magnitude of the induced torsion was designed to be the maximum allowable torsion (respecting safety margins) at the particular cross-section, rather then a possible bend-twist induced torsion magnitude. This was motivated by the idea of reducing any relative measurement error when proofing at the maximum allowable torsion, i. e., the maximum deformation.

Nevertheless, overall the aforementioned errors do not practically affect the validation of the model, as all DWS and load cables are modeled as LINK11 elements. With all attachment points modeled at their correct global location of the test setup. This assures that the forces and displacement measurement direction is always correct throughout the test, all under the assumption that the model behaves the same as the real blade. Thus, no corrections of any kind to measurements or FE results were applied.

## 4.4 Blade Segment Mass and Center of Gravity Measurement

After finishing the full blade tests, as discussed in Section 4.3, the blade was cut into 17 segments for further characterization. Figure 5 shows a cut surface of the 7[th] segment at a span-wise position of $r = 5.2$ m. To determine the 3D center of gravity (CoG), the segment was suspended at one point with a flexible rope, so that the CoG settled exactly underneath this point (like a pendulum). Hence, the vector in direction of the suspension rope defines an axis on which the CoG must be located (CoG axis). This procedure was repeated with different suspension points at least 2 times. The CoG was then found in the intersection point of the different CoG axes. The measurement setup can be seen in Fig. 10 as well as a digital representation of the intersection of different CoG axes.

To measure the vectors and analyze the data an optical measurement system (photogrammetry) was used. Every segment was equipped with several coded and uncoded reflecting marks to obtain the shape of the segment, the suspension points and a plummet that was used to get the CoG axes. All the point clouds were analyzed in Autodesk Inventor and Siemens NX. All segments were aligned in CAD and the CoG was extracted for each segment with regard to the blade coordinate system. In this way we obtained the distribution of the segment CoGs along the blade.

Considering the model validation, MoCA is able to generate the respective segments at their correct positions in the blade, so the segment masses and CoGs are a natural output of ANSYS.

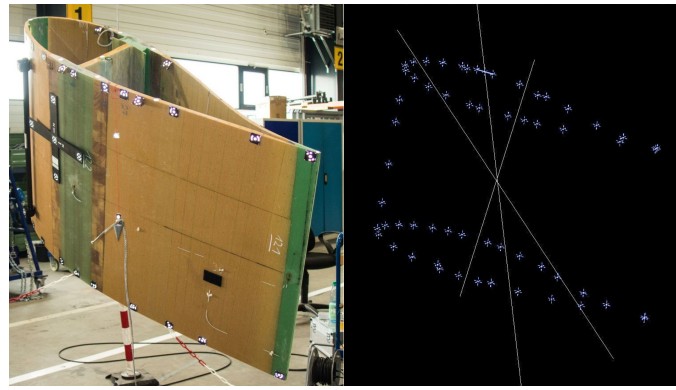

**Figure 10.** Measurement setup of cross-sectional analysis (left) and extracted vectors in CAD with intersection point defining the center of gravity (right).

## 5 Comparison of Experimental and Simulation Results

In this Sec., we compare the experimental results with the simulations. The observation scale will continuously decrease from a global to a more local scale. We start with the global blade characteristics such as natural frequencies, total mass, and global center of gravity. These give a rough estimate of the modeling correctness. Then the blade deformations by means of bending and twist distributions during the static extreme load tests will be analyzed. Finally, the strain levels in two cross-sections during the extreme load tests and the masses and centers of gravity of the cut blade segments are compared, which give a more detailed view on a local scale.

In order to rate the validity of the model, it is necessary to identify specific thresholds. However, these are hardly found in literature, especially as different applications and fidelity levels may demand other thresholds. Exemplary, Safarian (2015) reports validation requirements for finite element analysis according to the Federal Aviation Regulation of the U.S. government, where a displacement deviation of $< 5\%$ between the simulation and experiment is typically acceptable for global effects and local effects measured in form of strains allow for a maximum of $10\%$ deviation. Whereas strains exceeding these values require a re-evaluation of the model. These regulations refer to aviation applications, which also apply finite element shell models for the analysis comparable to our use case. Therefore, we will apply a $5\%$ threshold for global displacements, whereas a $10\%$ threshold will be applied on the cross-sectional strain results. These margins should also cover measurement uncertainties, as the DWS and the strain gauges offer a quite narrow uncertainty band, $0.6\%$ and $2\%$, respectively. Thresholds for masses are harder to define as these depend on the measurement setup, in our case with up to $2.5\%$ uncertainty. Plus, not all additional masses were correctly documented and thus not modeled. Same problem holds for natural frequencies $\omega$, following $\omega = \sqrt{\frac{k}{m}}$ and respecting unknown mass variation and typically a maximum of $5\%$ material tolerances (including density and stiffness according to private communication with manufacturers), it is also hard to define thresholds for the frequencies. Therefore, both mass and frequencies will be discussed individually.

## 5.1 Blade Mass, Center of Gravity, and Eigenfrequencies

Table 1 lists the total blade mass and the location of the center of gravity in longitudinal ($z$) and chord direction ($y$) as well as the measurement uncertainties and the deviation of the numerical model. We see that the model from MoCA is 115.5 kg lighter than the real blade, which corresponds to 6.44% relative difference related to the measurement. In contrast the measurement uncertainty is 45 kg. The mass difference is likely due to manufacturing deviations and/or additional masses (e.g., sensor wires and installations) that have not been considered in the numerical model. The location of the CoG matches perfectly in the chord direction, i. e., with a precision to the nearest two decimal places. There is only little deviation of 230 mm in the span-wise direction, which is almost within the measurement uncertainty range of $\pm$ 200 mm. All measurement uncertainties are based on given sensor uncertainties and taking the worst case scenario in the combination of those.

**Table 1.** Comparison of the total mass and the center of gravity (CoG).

|      | Experiment (in kg) | Uncertainty (in kg) | MoCA (in kg) | Difference (in kg) |
| ---- | ------------------ | ------------------- | ------------ | ------------------ |
| Mass | 1793               | 45                  | 1673.5       | -115.5             |

| CoG   | Experiment (in m) | Uncertainty (in m) | MoCA (in m) | Difference (in m) |
| ----- | ----------------- | ------------------ | ----------- | ----------------- |
| $y$   | 0.10              | 0.04               | 0.10        | 0.00              |
| $z$   | 6.58              | 0.20               | 6.35        | 0.23              |

The results of the modal analysis, both experimental and numerical, are listed in Table 2. The experimental results are taken from Gundlach and Govers (2019). The flapwise frequencies are in acceptable agreement with deviations of less than 8%. The largest deviation in flapwise modes is found for the $2^{\text{nd}}$ mode in the test rig configuration (7.94%, which corresponds to an absolute deviation of 0.54 Hz). The smallest deviation can be observed for the $1^{\text{st}}$ flapwise mode in the free-free configuration, which is 5.83% or 0.28 Hz, respectively. In edgewise direction, the approximation is even better. The largest relative deviation is seen for the $1^{\text{st}}$ edgewise mode in the test rig configuration, which is 4.84% (or 0.15 Hz in absolute numbers). The $2^{\text{nd}}$ edgewise mode is only 0.83% (or 0.09 Hz in absolute numbers) smaller in the simulation compared to the experiment in the test rig configuration, which is an excellent agreement. The largest absolute deviation is present in the free-free configuration, where the $1^{\text{st}}$ edgewise mode is 0.36 Hz lower than the measured value. Anyways, the deviation of the edgewise modes is less than 5% in all cases, which is a very good agreement. The $1^{\text{st}}$ torsion mode is quite well approximated in the free-free configuration, where the simulation is 5.62% lower than the experiment. However, in the test rig configuration the deviation is -11.76% (more than 2 Hz less compared to the test), which is relatively high. In general, the simulations agree better with the test results in the free-free configuration than in the test rig configuration. This is likely due to the rigid representation of the test rig and the connection bolts, as already mentioned in Sec. 4.2. Similar deviation ranges of the natural frquencies can be

found in Knebusch et al. (2020) for the same blade, but with a different model, with errors between 1.8% to 9.7% for flapwise
and edgewise modes and up to 22% for the torsion mode.

**Table 2.** Comparison of the modal analyses for the free-free (top) and the test rig (bottom) configuration. Experimental results are taken from (Gundlach and Govers, 2019).

| Mode free-free | Experiment (in Hz) | MoCA (in Hz) | Difference (in Hz) | Difference (in %) |
|---|---|---|---|---|
| 1st flapwise | 4.8 | 5.08 | 0.28 | 5.83% |
| 1st edgewise | 10.1 | 9.74 | -0.36 | -3.56% |
| 1st torsion | 16.9 | 15.95 | -0.95 | -5.62% |

| Mode test rig | Experiment (in Hz) | MoCA (in Hz) | Difference (in Hz) | Difference (in %) |
|---|---|---|---|---|
| 1st flapwise | 2.2 | 2.37 | 0.17 | 7.73% |
| 2nd flapwise | 6.8 | 7.34 | 0.54 | 7.94% |
| 1st edgewise | 3.1 | 3.25 | 0.15 | 4.84% |
| 2nd edgewise | 10.9 | 10.81 | -0.09 | -0.83% |
| 1st torsion | 18.7 | 16.50 | -2.20 | -11.76% |

## 5.2  Static Bending Tests

The results of the static bending tests will be illustrated by means of deflection lines. For each test setup, two lines exist, one for the front and one for the rear DWS. The deflections in the front DWS are plotted in Fig. 11 for each pausing load during ramp-up (40%, 60%, 80% and 100% of the target load as described in Sec. 4.3). The plots for the rear DWS are added in appendix A.
A Table is added in each of the Figures that show the differences between the simulations and the tests (in absolute and relative numbers). The tip DWS values are the same for the rear and the front DWS, as only one DWS is installed at the blade tip.

Figure 11 (a) shows the result of the front DWS during the MXMAX load case. For this scenario a maximum deflection of 180 mm at the blade tip is reached. The simulation shows excellent agreement for the front DWS sensors, with a maximum absolute difference of -2.3 mm at the tip for 100% load and a maximum relative difference of -4.0 at LF1, whereas the
deviations in all other positions are well below 2.0%. The rear DWS results shown in Fig. A1 (a) in the appendix A have slightly higher errors with a maximum of -5.5% at LF1 for full load.

For the load case MXMIN, Fig. 11 (b) illustrates the front DWS results. Except for LF1 the results are in very good agreement with a maximum deflection error of -1.6% at LF2 at full load. However, the results in LF1 return maximum errors of 3.8% at

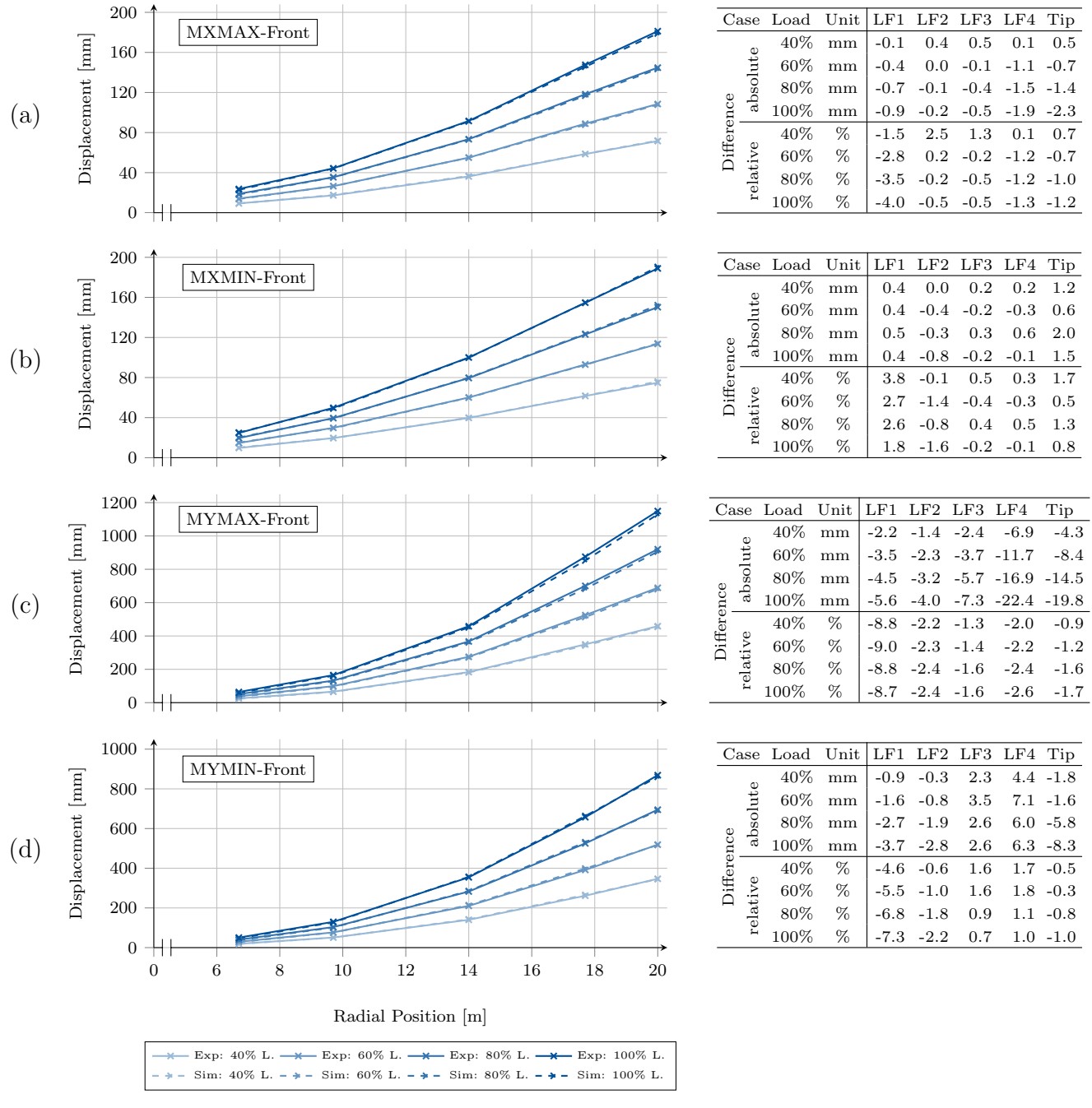

**Figure 11.** Bending lines extracted from the front draw wire sensor for the (a) MXMAX; (b) MXMIN; (c) MYMAX; (d) MYMIN experiment and simulation. Results are shown for 40%, 60%, 80% and 100% of the target load. The Table on the right shows the differences between the simulation and the test.

40% load, which decreases to 1.8% at full load. Similar behavior is found for the rear DWS (Figure A1 (b)); excluding LF1 the maximum error is 1.7% in LF3 and the tip during 40% load.

The results of the front DWS during the maximum flapwise setup (MYMAX, Fig. 11 (c)) are in very good agreement, when excluding the LF1 data. The LF1 results tend to show the highest errors. This might probably be due to the smallest absolute deflection values, as a systematic sensor/measurement inaccuracy will have a higher impact on relative errors. Concerning the other load frames the maximum error is found to be -2.6% for the LF4 DWS at full load, which corresponds to -22.4 mm

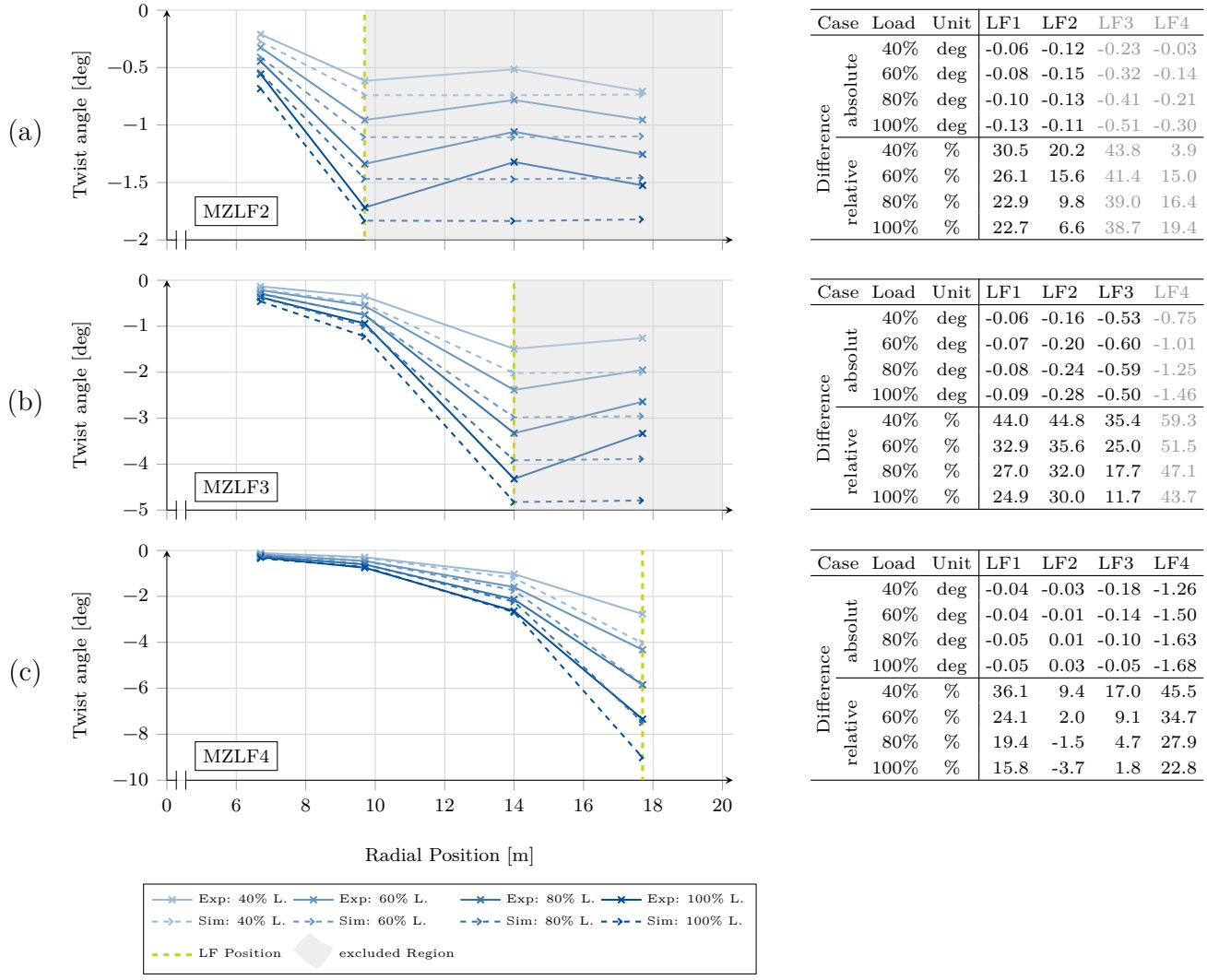

**Figure 12.** Twist angles calculated from the draw wire sensors results for the (a) LF2; (b) LF3; (c) LF4 torsional loading experiment and simulation. Results are shown for 40%, 60%, 80% and 100% of the target load. The Table on the right shows the differences between the simulation and the test.

deflection error at a maximum deflection of 875 mm in the experiment. All other values range between -0.9% and -2.4%. The excluded LF1 results show higher errors of up to 9.0% for 60% load. For the rear DWS (Figure A1 (c)), though excluding LF1 (max. error -17.6%), the LF2 results show errors above 6.7% with the highest reaching -8.8% during full load. For the other two load frames the errors are low again and are between -0.9% and -2.4%. If taking a closer look at the LF2 full load deflection $d$ in the test and experiment the front DWS shows $d_{Exp,f} = 165$ mm and $d_{Sim,f} = 161$ mm, whereas the rear DWS returns

$d_{Exp,r} = 175$ mm and $d_{Sim,r} = 160$ mm. That means the overall deflection of the simulation is less than in the experiment, but the difference between rear and front is $\Delta d_{Exp} = d_{Exp,r} - d_{Exp,f} = 10$ mm and $\Delta d_{Sim} = -1$ mm, i. e., the simulation shows a positive rotation around the z-axis, while the experiment returns a much higher negative rotation. The rotation angle $\Theta$ can be calculated by the relationship

$$\Theta = arcsin\left(\frac{\Delta d}{l_{SP}}\right), \tag{1}$$

where $l_{SP}$ is the distance of both front and rear DWS attachment points on the load frame. The rotation angle becomes $\Theta_{Exp,LF2} = -0.268°$ in the experiment and $\Theta_{Sim,LF2} = 0.042°$ in the simulation. Assuming the pivot point is at the shear center (SC), a correction could be calculated to see if the bad results of the rear DWS at LF2 is due to the wrong predicted rotation along the z-axis. All necessary geometric data can be found in Haller and Noever-Castelos (2021). Following Eq. (1) and using the distance of the front or rear DWS attachment to the shear center, the front absolute difference is increased to

-7.87 mm which results in an error of -4.8% and the rear deflection is reduced to -7.7 mm, respective -4.4% during 100% loading. By this correction due to a wrong predicted rotation angle the rear DWS approximation improves by 4%, while the accuracy of the front sensor decreases by only 2.4%. This correction is introduced to evaluate the accuracy for the bending prediction and only holds for the LF2 position, as the other positions have different rotation angle deviations. Additionally, it has to be noted that during the flapwise loading the DWS attachment distances to the shear center are much higher than for the

edgewise loading, i. e., the influence from rotation angle deviations is amplified significantly.

Figure 11 (d) shows the front DWS results comparison during the minimum flapwise loading scenario (MYMIN). All load frames are installed so can be evaluated and the results show a very good agreement with errors below 2.2% for all DWS, except LF1. At this first load frame, again the results have significantly higher errors of up to -7.3% at full load. Figure A1 (d) contains the rear DWS result of the MYMIN load case and lists throughout higher deviations of up to -13% for the LF1 sensor. Here

again, by analyzing the rotation behavior of the blade along the z-axis all load frames show significant rotation differences and after estimating a correction, e.g., the accuracy of the LF1 front sensor would decrease to a deviation of -11%, while that of the rear sensor increases to -10.4%. This is the worst approximation of the simulation for the static extreme load bending setups. The other load frames are in very good agreement, with most of the results (Excluding LF1) being below the 5% threshold defined at the beginning of Sec. 5.

## 5.3  Static Torsion Tests

Full scale blade tests in pure torsion are usually not included in certification processes according to (International Eletrotechnical Comission, 2014) and are thus rarely available. As described in Sec. 4.3 the blade is twisted during three different setups

successively at the load frames LF2, LF3 and LF4. The results of the tests and the simulations are plotted in Fig. 12. The structural behavior behind the actual loaded frame position to the tip will not be addressed in this paper and is highlighted as grey-colored areas, as the areas loaded in torsion are located between the root and the respective load frame. However, the raw results similar to the static bending experiments are the DWS length variation, for these torsional experiments the more relevant twist angles are calculated and plotted according to Eq. 1. Figure 12 (a) shows the first torsional test loaded at LF2. The absolute angle deviation from experiment to simulation are in between -0.06° and -0.15° but yields high relative deviation up to 30% due to the small twist angles of -0.55° at LF1 and -1.72° at LF2 during 100% load.

Moving the load application to LF3 (Figure 12 (b)) does not change the situation. At the load application position the absolute error is high with up to -0.6° at a maximum twisting of -4.3°. All errors exceed -10% dramatically. However, the experiment with torsional loading on LF4, see Fig. 12 (c), shows reasonably good results for the twist angle at LF2 and LF3 with angle deviations of 3.7% and 1.8%, respectively. The results at LF4, where the load is applied and which shows the highest twist angle keeps high deviations of about 20% for full load. Such high errors during torsional loading may base on the shell element with a node offset to the exterior surface used for this model. Greaves and Langston (2021), Branner et al. (2007), Pardo and Branner (2005) and especially Laird et al. (2005) already stressed the high inaccuracy of shell elements with node offsets from the mid-plane to predict the structural behavior of hollow structures subjected to torsional loading. However, the twisting is generally overestimated throughout the three torsional tests, which is inline with the aforementioned references.

## 5.4 Local Strain Comparison

As stated in Sec. 4.3 the highly instrumented cross-sections at $r = 5$ m and $r = 8$ m offer a more detailed view on the local strain levels in the rotor blade. The strain results are used to compare the simulations with the tests and to verify that local effects are correctly reproduced. We have selected a few representative load cases in this section. The remaining load cases can be found in appendix B.

In Fig. 13 (a) the strain in 0° (span-wise direction, in blue) and 90° (cross-wise, in yellow) directions for the MXMIN simulation (solid lines) and experiment (circles) are plotted over the normalized airfoil circumference (denoted by $S$) for the 5 m cross-section, starting at the suction side trailing edge ($S = 0$), moving along the suction side to the leading edge ($S \approx 0.5$), and then along the pressure side to the pressure side trailing edge ($S = 1$). This cross-section shows some general characteristics in all load cases, which are:

1. In the simulation at $S = 0$, there is a strain peak in the 90° direction, because the sandwich core material vanishes suddenly towards the trailing edge, due to the shell elements and their missing capability of tapering single materials in their sections as done in the real layup.
2. In the simulation at $S = 0 - 0.25$, there is an excessive or wrong curvature in the 90° strain curve, for which we do not have a feasible explanation.
3. In the simulation at $S = 0.25 - 0.35$, there is a stepped dip or raise of the 90° strain, because the sandwich core material is substituted by core and UD layers and then completely by the UD spar cap and vice versa.

4. In the simulation at $S = 0.5$, there is a strain peak in the 90° direction, because the sandwich core material vanishes around the leading edge.

5. In the experiment at $S = 0.5 - 0.65$, there is a strain dip in the 0° direction, for which we do not have a feasible explanation. The structure should be symmetric next to the leading edge.

6. In the simulation at $S = 1$, there is a strain peak in the 90° direction, because the sandwich core material vanishes towards the trailing edge.

Apart from the unclear dip around the suction side leading edge panel ($S = 0.5 - 0.65$), the longitudinal strain (in 0° direction) differs along the circumference in mean only about $\pm 107 \ \mu m/m$. This is about 13.3% related to the maximum measured absolute strain of $811 \ \mu m/m$. However, the cross-wise strains (in 90° direction) reach deviations of up to $\pm 90 \ \mu m/m$ in mean, which corresponds to about 26.2% related to its measured maximum. The MXMAX results (Figure B2 (a)) are slightly better concerning mean strain errors, with 12.4% for the 0° direction and 20.3% for the 90° direction.

Figure 13 (b) shows the MYMAX load case. Unlike the edgewise case a failure of the strain gauge at $S = 0.3$ was recorded in the experiment, which can be seen in the discontinuity of the experimental results. The flapwise bending of the blade in general is more excessive compared to the edgewise bending and provokes the highest longitudinal strains in the spar cap positions reaching maximum values of up to $1800 \ \mu m/m$ in the outer shell layer. Consequently the cross-wise strain also increases with absolute mean errors to $\pm 208 \ \mu m/m$ (11.6%) in 0° direction and $\pm 217 \ \mu m/m$ (36.0%) in 90° direction, both approximately twice as much as in the edgewise load case. All other aforementioned issues are also present here, some more and some less pronounced. The same conclusion also holds for the MYMIN case in Fig. B2 (b), though the mean error values are lower, due to smaller load sets. In 0° direction a mean error 8.2% was calculated and 15.8% in 90° direction.

Taking a look at the torsion tests, in particular for the MZLF3 load case plotted in Fig. 13 (c), the longitudinal strain shows a relatively good agreement with the test, except for $S = 0.5 - 0.65$ and at the pressure side trailing edge panel ($S = 0.85 - 1$). The cross-wise strain shows partially good agreement with the experiments, except for the aforementioned characteristics which are more dominant than in the bending tests, e. g., the peaks at the trailing edge is more pronounced. As for the longitudinal strain, the cross-wise strain shows a disagreement between simulation and experimental results, which is even stronger due to a shifted curvature in the plot. These can also be seen during the remaining two torsion tests. The MZLF4 load case in Fig. B2 (d) is very similar to the MZLF3 load case, whereas the MZLF2 load case (Figure B2 (c)) shows all of the stated characteristics in a more pronounced manner as the load introduction is shifted closer to the evaluated section at $r = 5$ m.

The next highly equipped cross-section is at $r = 8$ m. While the previous cross-section was located at maximum chord, this one is already in a region where geometric curvatures are smoother. For direct comparison the same three load cases were selected for this cross-section. As depicted in Fig. 13 (d) the longitudinal as well as the cross-wise strains during the MXMIN test follow very well the experimental results, both qualitatively and quantitatively. Strain levels are similar to the cross-section at $r = 5$ m, but the strain errors of the simulation compared to the experiments are much lower (mean error $\pm 29 \ \mu m/m$ or 3.0% in 0° direction and $\pm 32 \ \mu m/m$ or 11.0% in 0° direction ). Same holds for the MXMAX loading, see Fig. B4 (a), where the mean strain error is even between 2.0% and 9.5%, respectively. Although these are not very pronounced, the peaks at the trailing and leading edges as well as the stepped dips or raises can be identified as consistent characteristics throughout all test.

Comparing the results of the MYMAX test depicted in Fig. 13 (e), the good agreement between the simulation and the test are evident. Even the stepped raise at the two spar caps ($S = 0.3$ and $S = 0.67$) exist in the experimental results. The mean strain error is $\pm 53\,\mu m/m$ (2.5%) in 0° direction and $\pm 63\,\mu m/m$ (8.4%) in 90° direction, which is much less than for the other cross-section, while having slightly higher maximum strain levels of $2080\,\mu m/m$ in 0° direction and $753\,\mu m/m$ in 90° direction. This excellent agreement is also found in Fig. B4 (b) for the MYMIN load case.

However, the results from the torsional tests do not agree. As seen in Fig. 13 (f) the simulation results of the longitudinal strain during the MZLF3 test may follow some correct trend of the experiments, but has significant differences. The same applies to the cross-wise strains. Although the strain errors are in the same range as the bending test results, compared to the absolute strain levels these have the same magnitude as the error. The remaining torsional test results (Figure B4 (c) and (d)) show similar problems.

## 5.5  Segment mass and CoG comparison

In this subsection, we compare the experimental mass and CoG measurement of each segment with the respective simulation results. Table 3 contains the segment numbers, the segment locations along the blade defined by the span-wise positions of the left and the right cutting sections $r_1$ and $r_2$, respectively, and the differences of the segment masses and the CoG locations (in absolute and relative numbers).

The relative difference of the mass is related to the measured segment mass and the CoG positions are with respect to the corresponding geometrical mid cross-sectional dimensions, i. e., absolute thickness (X), chord length (Y) and radial segment length(Z). It was not possible to measure segment 15. The mass differs from -4.8% to -9.9% except for segment 1,14, and 17, where the mass was overestimated. Unfortunately it was not possible to calculate an overall blade mass as one segment result was missing. Concerning the CoG differences, the coordinate in cross-section thickness direction (X) varied up to -15% but was most of the time predicted closer to the suction side. The CoG location in chord direction (Y) agreed very well with the measurement, except for segment 16, the variation were below ±4%. The radial locations match well for most of the segments ($\leq$ 6%). However, the sections 10, 11, 12, 14, and 16 resulted in higher variations, predicting the CoG position closer to the tip by more than 10% of the segment length.

## 6  Summary & Conclusion

The aim of this paper was the validation of a parameterization and modeling methodology for wind turbine rotor blades. This methodology was implemented in the in-house 3D finite element model generator MoCA (Model creation and analysis tool), which creates hybrid shell/solid finite element models.

Full-scale blade tests were performed on the SmartBlades DemoBlade as an experimental reference. The blade has a length of 20 m and is designed with pre-bend and pre-sweep. The following magnitudes were determined experimentally: The total mass and the center of gravity of the full blade, the mass and center of gravity distributions along the blade by weighing of

blade segments, the natural frequencies in a free-free and a clamped cantilever configuration, the deflection curves along the blade for both flapwise and edgewise bending as well as torsion, and the strains in the cross-sectional and longitudinal direction close to the maximum chord position. The governing parameters such as geometry, material layup, manufacturing deviations, additional sensor and load frame masses were extracted from the blade and test documentations. These were fed into MoCA. Finite element models for all test setups were created and the simulations were executed in the commercially available finite element code ANSYS. Then, the simulations were compared with the experimental results.

The mass and center of gravity of the full blade compared very well (error of -6%). The masses and centers of gravity of the blade segments, i. e., the mass and center of gravity distributions along the blade, were also in good agreement (error of 5-10%). Modal analysis concluded for the natural frequencies with free-free boundary conditions also well (error <6%) matching results, those for the clamped cantilever configuration matched reasonably well (error <8% for bending, 11.7% for torsion). The deflections for bending in edgewise direction was in excellent agreement (error <4%). While the deflection curve for bending in flapwise direction showed a comparably large deviation of 13% at the root, which decreased substantially towards the tip (error at the tip <4%). A reason for that was an elastic twist during the test that was not replicated in the simulations. In general, the errors mostly comply with the validation threshold 5% defined at the beginning.

For both flapwise and edgewise bending the strains in the span-wise direction were in reasonable good agreement, taking into account, that no local mesh refinement or global-local-modeling strategy was followed. Strain gauges were distributed along the circumference of the cross-sections at span-wise positions of 5 m and 8 m, respectively, in order to measure the cross-sectional deformations. Especially at a span of 8 m, the authors observed a very good agreement of the simulation and the experiments, with nearly all mean strain errors below the 10% threshold defined for local comparison. The cross-section at a span of 5 m produced errors approximately at twice the errors of the 8 m section. However, for both sections some local effects close to the spar-caps could not be resolved in the simulations.

During torsion, the authors identified quite large deviations in the global elastic twist distributions along the blade. Also the 1st torsional natural frequency has the highest discrepancy to the test with -11.76%. During the torsion test the strain measurements showed quite large deviations exceeding 30% mean errors at 8 m span and reaching up to 295% mean error at 5 m span. Though, the longitudinal strains agreed better than the transverse strains, at least qualitatively. As literature reports, all this may by traced back to the shell elements being inappropriate to model torsional behavior, due to the offset of the nodes to the element's mid-plane.

Generally speaking, the authors observed good agreement between the simulations and the experiments in almost all situations global bending observations and acceptable agreement in local observations. The parameterization and modeling methodology can thus be rated as validated, in the capabilities of the proposed modeling technique.

However, modern flexible blade design, which is driven to it's material and structural integrity limits and includes intentional torsion for load alleviation, requires accurate predictions for all load cases in order to be reliable. Looking a step further,

especially fatigue damage calculation, needs correct strain or stress predictions of the models. The authors currently work on evaluating blade modeling by means of solid elements and/or solid shell elements. Although, we loose computational efficiency of the shell element models, this way, the accuracy in torsional response should be improved significantly. Additionally, the correct representation of geometrical shape and 3D tapering can be realized. This should shed light on the discrepancy in torsion and some of the bending load cases, where we were unable to identify their origin, for instance wrong curvatures in the strain distributions or numerical steps/peaks at material tapering. However, such very local effects as material discontinuities and numerical strain/stress peaks, probably require a global-local modeling approach to capture every smaller scaled detail. Subsequently, a sensitivity study of relevant geometry, material or modeling parameters can enhance further the understanding of local inaccuracies.

*Code and data availability.* The code of MoCA is not publicly available, but may be made available on request at conditions that need to be agreed upon. All experimental and simulation data that support the results of this research as well as the baseline finite element model of the blade as an ANSYS mechanical input file are uploaded in Noever-Castelos et al. (2021)

**Appendix A: Static bending test results**

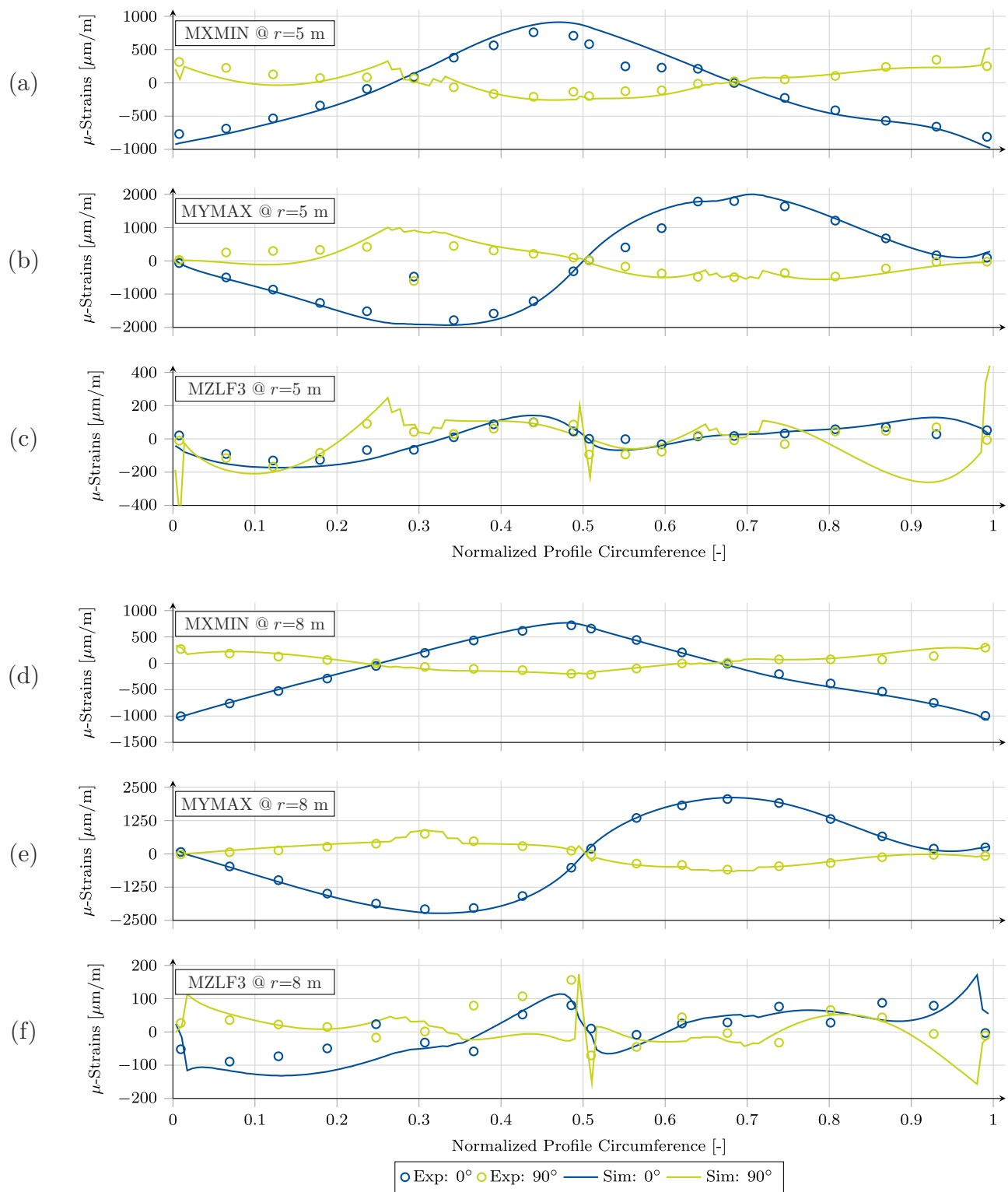

**Figure 13.** Span-wise and cross-wise strains of the simulation and the test, plotted against the normalized profile circumference of the cross-section at $r$ = 5 m for the (a) MXMIN; (b) MYMAX; (c) MZLF3 load case and at $r$ = 8 m for the (d) MXMIN; (e) MYMAX; (f) MZLF3 load case.

**Table 3.** Segment mass and center of gravity (CoG) differences between experiment and simulation. The relative distances of the CoG are given with respect to their corresponding geometrical cross-section parameter, i. e., absolute thickness (X), chord length (Y), and span-wise segment length (Z).

| Section No. | $r_1$ (in m) | $r_2$ (in m) | Mass (in kg) | Mass (in %) | Center of Gravity X (in m) | Y (in m) | Z (in m) | X (in %) | Y (in %) | Z (in %) |
|---|---|---|---|---|---|---|---|---|---|---|
| 1 | 0.0 | 0.9 | 34.6 | 9.8% | -0.030 | 0.000 | 0.003 | -2% | 0% | 0% |
| 2 | 0.9 | 2.0 | -7.36 | -5.1% | -0.003 | 0.009 | 0.035 | 0% | 1% | 3% |
| 3 | 2.0 | 3.0 | -10.96 | -9.3% | -0.031 | -0.004 | 0.065 | -3% | 0% | 6% |
| 4 | 3.0 | 3.5 | -4.74 | -8.0% | -0.066 | 0.000 | -0.007 | -6% | 0% | -1% |
| 5 | 3.5 | 4.0 | -3.419 | -6.1% | -0.076 | -0.005 | 0.021 | -8% | 0% | 4% |
| 6 | 4.0 | 5.2 | -7.39 | -5.9% | -0.094 | -0.060 | 0.055 | -10% | -3% | 5% |
| 7 | 5.2 | 6.5 | -6.07 | -4.9% | -0.102 | -0.036 | 0.054 | -13% | -2% | 4% |
| 8 | 6.5 | 8.5 | -9.81 | -5.8% | -0.074 | -0.008 | 0.071 | -12% | 0% | 4% |
| 9 | 8.5 | 9.5 | -3.572 | -4.8% | -0.050 | 0.007 | 0.040 | -10% | 0% | 4% |
| 10 | 9.5 | 10.5 | -5.236 | -7.3% | -0.049 | 0.004 | 0.132 | -11% | 0% | 13% |
| 11 | 10.5 | 11.5 | -3.685 | -5.4% | -0.041 | -0.005 | 0.108 | -10% | 0% | 11% |
| 12 | 11.5 | 12.5 | -4.087 | -6.6% | -0.031 | 0.003 | 0.090 | -9% | 0% | 9% |
| 13 | 12.5 | 16.0 | -18.59 | -9.9% | -0.036 | 0.007 | 0.091 | -13% | 1% | 3% |
| 14 | 16.0 | 16.5 | 4.007 | 16.3% | -0.003 | -0.048 | 0.128 | -1% | -4% | 26% |
| 15 | 16.5 | 17.5 | | | | | | | | |
| 16 | 17.5 | 19.0 | -4.405 | -9.1% | -0.025 | 0.094 | 0.195 | -15% | 11% | 13% |
| 17 | 19.0 | 20.0 | 1.104 | 9.5% | 0.010 | 0.023 | 0.041 | 10% | 4% | 4% |

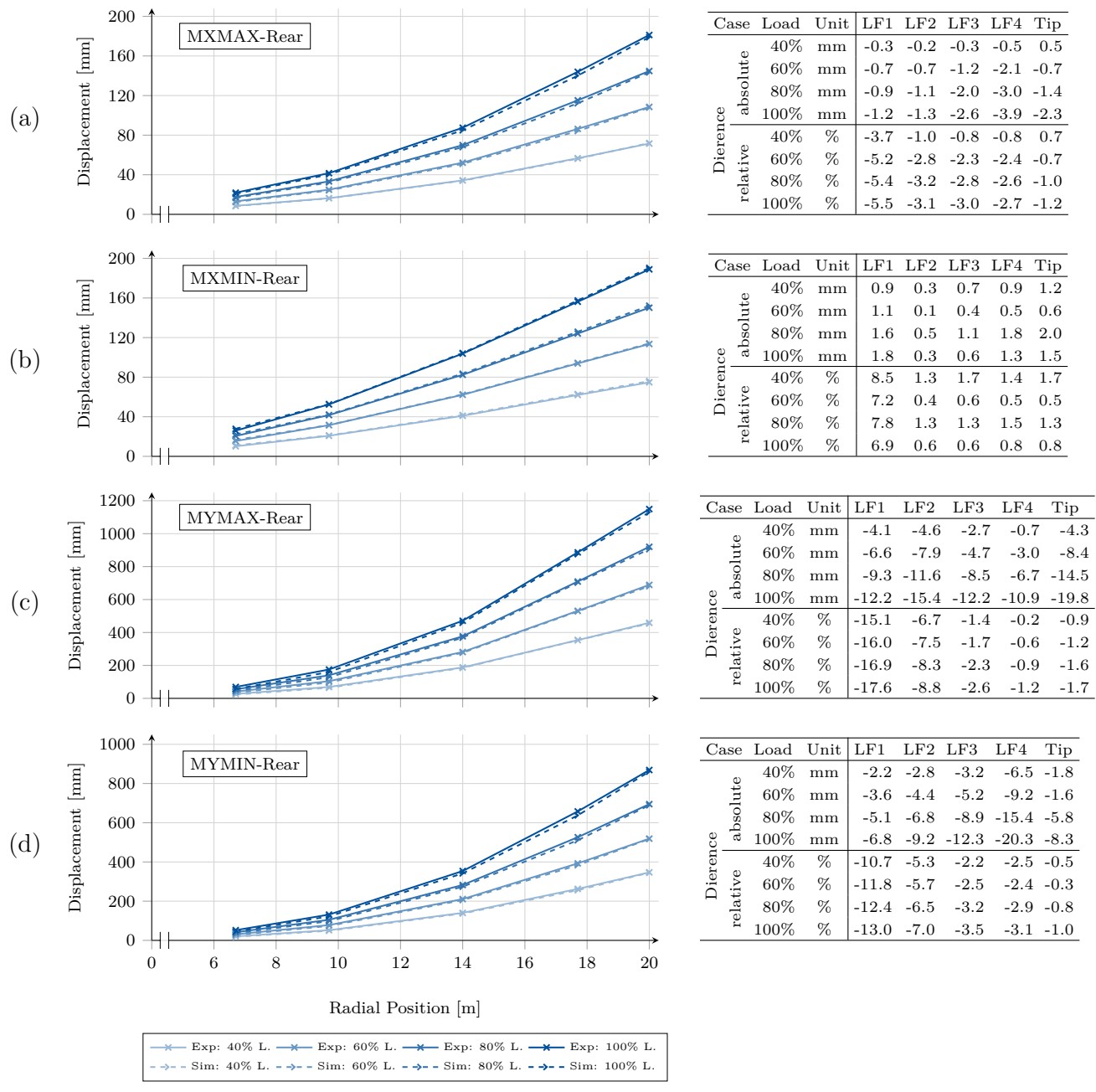

**Figure A1.** Bending lines extracted from the rear draw wire sensor for the (a) MXMAX; (b) MXMIN; (c) MYMAX; (d) MYMIN experiment and simulation. Results are shown for 40%, 60%, 80% and 100% of the target load. The Table on the right shows the differences between the simulation and the test.

# Appendix B: Local strain comparison

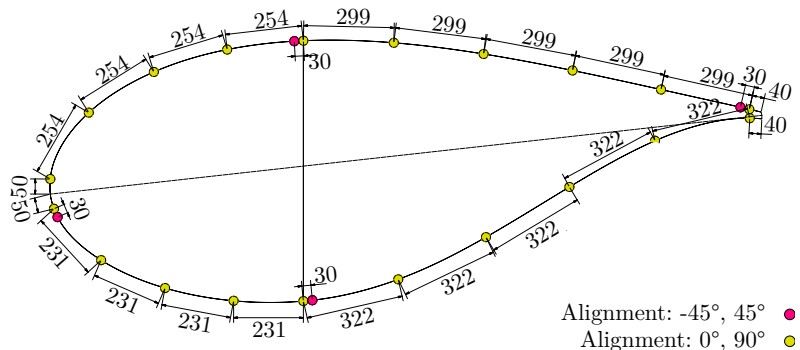

**Figure B1.** Cross-sectional sensor distribution at $r = 5$ m ($r_{\text{norm}} = 25\%$) (Haller and Noever-Castelos, 2021)

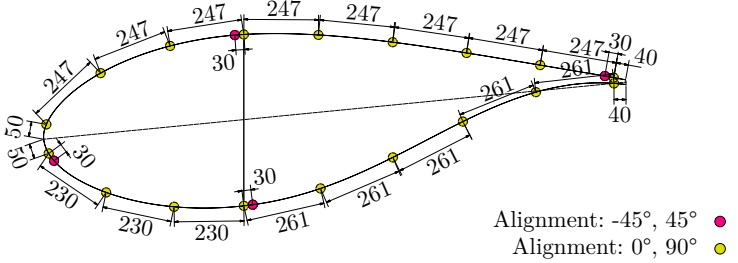

**Figure B2.** Span-wise and cross-wise strains of the simulation and the test, plotted against the normalized profile circumference of the cross-section at $r = 5$ m for the (a) MXMAX; (b) MYMIN; (c) MZLF2; (d) MZLF4 load case.

**Figure B3.** Cross-sectional sensor distribution at $r = 8$ m ($r_{norm} = 40\%$) (Haller and Noever-Castelos, 2021)

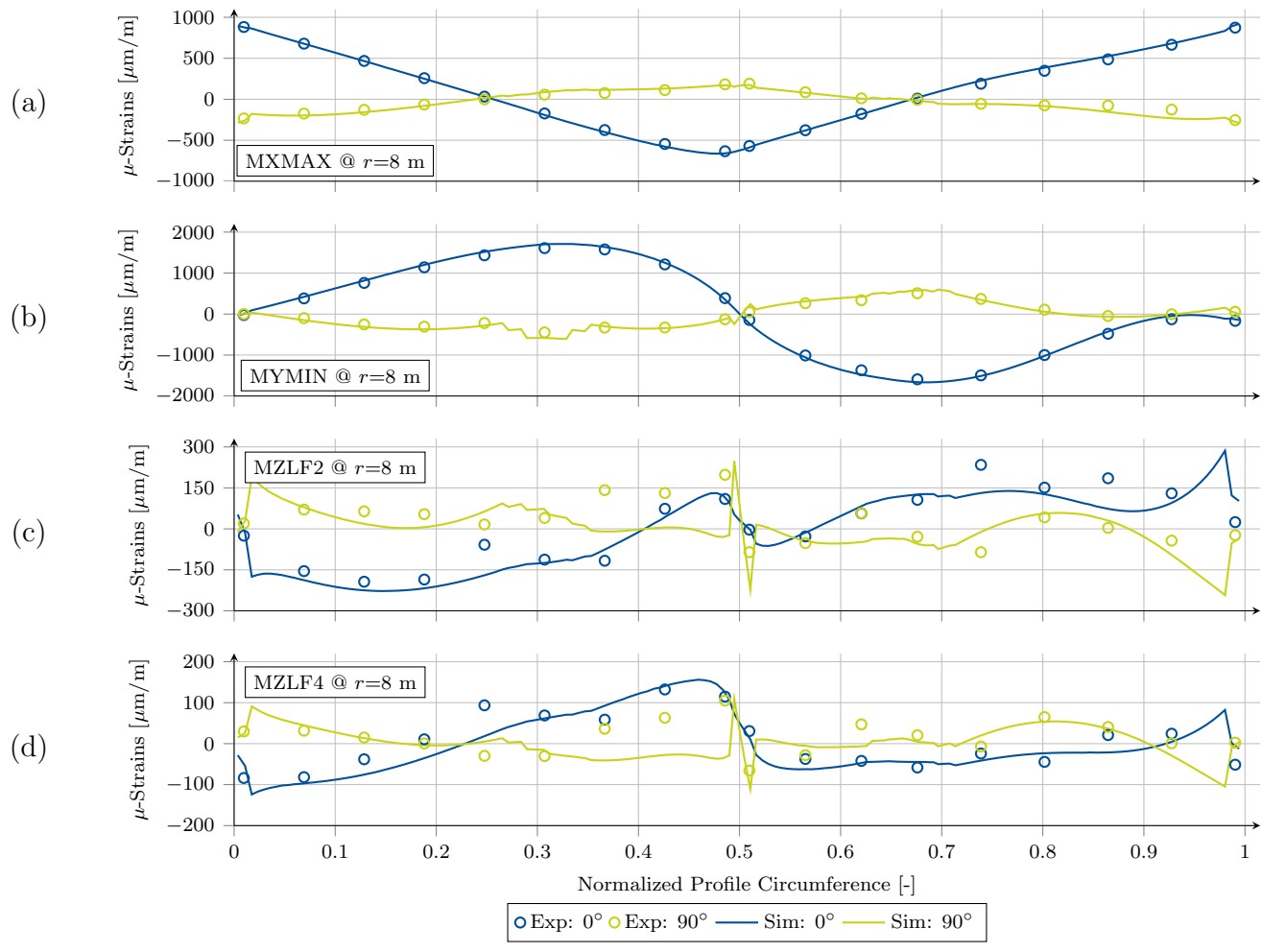

**Figure B4.** Span-wise and cross-wise strains of the simulation and the test, plotted against the normalized profile circumference of the cross-section at $r = 8$ m for the (a) MXMAX; (b) MYMIN; (c) MZLF2; (d) MZLF4 load case.

*Author contributions.* Pablo Noever-Castelos implemented the parametrization and modeling methodology in MoCA, conducted the numerical simulations, compared the simulations with the tests, and wrote the paper. Bernd Haller planned, executed, and documented the tests. Claudio Balzani is the supervisor and guided Pablo Noever-Castelos in the conception of the ideas and participated in the specification of strain gauge positions as well as in writing, structuring, and reviewing the paper.


*Competing interests.* The authors declare that they do not have any conflicts of interests.

*Disclaimer.* The information in this paper is provided as is and no guarantee or warranty is given that the information is fit for any particular purpose. The user thereof uses the information at its sole risk and liability.

*Acknowledgements.* The authors acknowledge the financial support by the Federal Ministry for Economic Affairs and Energy of Germany in the project *SmartBlades2* (project numbers 0324032B/C). The authors further acknowledge the coordination effort of the German Aerospace Center (DLR), the very good cooperation with the project partners and the fruitful discussions within the project consortium.

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
