# Peer review of "Validation of a modeling methodology for wind turbine rotor blades based on a full scale blade test"

_Wind Energy Science, 2021_

## Author Response (AR1)

*Dear Martin Eder,*

*Thank you for taking the time to study this paper and provide valuable constructive Criticism, which we believe has helped develop and strengthen this work significantly. Please find our answers to all your comments, by either corrected or added text sections or comments on your suggestions/concerns. We hope to have adequately accommodated all your concerns and thank you again for your contribution.*

**General comments:**

- **The introduction should contain a short paragraph (e.g. bullet point format) highlighting both the novelty and the significance of the proposed approach compared to readily established classic approaches.**

  *The introduction was rearranged into subsections. Subsection '1.2 Objectives of this paper' contains the following paragraphs:*

  *"Though some of these model creation frameworks may work with functions or splines describing the blade's geometrical or layup information, most of them work with a reasonably high number of airfoils/stations that in addition to the blade's geometry yield the outer blade shape by a global linear or higher order interpolation between the airfoils.*

  *The presented method combines and extends several aspects of the different aforementioned software packages. The benefits are:*

  - *it generates airfoils independent from any neighbouring geometry and uses the relative thickness distribution to position these along the span. This assures the geometry distribution, as this avoids any overshoot due to spanwise geometry interpolation.*
  - *any parameter which may vary over the radius can be defined as spline, e.g., relative blade thickness, layer thickness, material density or stiffness.*
  - *it enables flexible and easy parameter studies due to the simple parameter variation based on splines.*
  - *it is designed for research, as different modules can be easily replaced by an alternative code, e.g., airfoil interpolation, adhesive modeling.*
  - *it generates an FE-Model in MATLAB and provides already an interface to ANSYS APDL and BECAS, however interfaces to other FE-software can easily be implemented."*

- **The authors conclude that shell elements do not lend themselves to predicting the torsional twist distribution along the blade. This problem is well-known to persist (also) in blade models and has been previously reported in the literature by many different researchers (see attached literature for inspiration). The problem is related to the midplane offset function and the integration scheme of the shell elements. The offset is used to form a smooth outer surface defined by the airfoils.**

  *This problem is already discussed at the end of Section 5.3. It was extended by two further references:*

  *"Such high errors during torsional loading may base on the shell element with a node offset to the exterior surface used for this model. Greaves.2021, Branner.2007, Pardo.2005 and especially Laird.2005 already stressed the high*

*inaccuracy of shell elements with node offsets from mid-plane to predict the structural behaviour of hollow structures subjected to torsional loading. However, the twisting is generally overestimated throughout the three torsional tests, which is inline with the aforementioned references."*

- **It is commonly accepted that the accuracy of the torsional stiffness prediction can significantly be improved when adopting layered solid elements or continuum shell elements. This solution of course comes at a price of computational efficiency. However, the pros and cons of this solution should be highlighted in the conclusions.**

**&**

- **The authors state that the measured and predicted strain distribution in their torsion tests is largely at variance. This statement is actually quite significant and maybe alarming: numerical fatigue damage prediction, continuum damage models, equivalent strain envelope for fatigue tests to name a few. The authors should provide a more thorough explanation for possible root causes. Is it deemed to be a measurement error or rather a simulation error (i.e. modelling artefact)? Did the predicted strains improve when using a different element type?**

*We fully agree to this statement and the necessity with concerns to:*

*"However, modern flexible blade design, which is driven to it's material and structural integrity limits and includes intentional torsion for load alleviation, requires accurate predictions for all load cases in order to be reliable. Looking a step further, especially fatigue damage calculation, therefore, needs correct strain or stress predictions of the models."*

*This is specially important, with respect to the error in torsional prediction.*

*"As literature reports, all this may be traced back to the shell elements being inappropriate to model torsional behaviour, due to the offset of the nodes to the element's mid-plane."*

*And this is why:*

*"The authors currently work on evaluating blade modeling by means of solid elements and/or solid shell elements. Although, we loose computational efficiency of the shell element models, this way, the accuracy in torsional response should be improved significantly, as well as the correct representation of geometrical shape and 3D tapering can be realized. This should shed light on the discrepancy in torsion and some of the bending load cases, where we were unable to identify their origin, for instance wrong curvatures in the strain distributions or numerical steps/peaks at material tapering. However, such very local effects as material discontinuities and numerical strain/stress peaks, probably require a global-local approach to capture every smaller scaled detail. Subsequently, a sensitivity study of relevant geometry, material or modeling parameters can enhance further the understanding of local inaccuracies."*

- **The authors have measured the torsional twist through application of a force couple on the saddle or yoke mounted on the blade. It is not entirely clear how the sway induced by the bend-twist coupling was considered in the measurement. Secondly, the bend twist coupling response might entail a rotation of the cross section closer to one (or at**

**one) of the two loading points rather than the centre. Moreover, if the load was applied through cables, the bend-twist coupling effect might have changed the direction of the two force vectors. The authors should provide more detailed information regarding the test and measurement procedure. One (maybe of many) alternative testing method could have been to attach a transverse lever to the yoke. One test LC1 is performed by conducting a pure flapwise test by loading the yoke centrally. A second test LC2 is performed by a combined flapwise-torsion test by loading the yoke at the tip of the lever. Using the superposition principle, the torsional twist can be found by subtracting LC2 from LC1. However, the authors should explicate their reason for choosing their method and why they consider it superior to other existing approaches together with possible limitations of their approach.**

*We hope the following paragraphs describes the torsional test accurately enough and give an explanation why we have chosen this method:*

*"Because the blade is still mounted at a block angle of 7.5° the torsional moment is not fully aligned with the pitch axis, as the forces do not act exactly in the cross sectional plane. The load cable oriented upwards was attached to a ceiling crane and to the load frame at approximately the shear centre position. As the ceiling crane location is hard to record, but the load rope is perpendicular to the ground it was assumed that the location is 18m above the corresponding load point ($y$-direction, approximately crane height). By this, deflections parallel to the floor, due to load application, would only result in small angle deviation of the perpendicular force. The force facing downwards was applied onto the load frame corner to create the lever with respect to the shear center. Our procedure is similar to a combination of the pure torsion and locked torsion test presented by Berring.2007. However, this method may imply some errors from:*
- *numerical shear center calculation*
- *not suspending exactly at the cross sectional shear center, but on the frame, which leads to an offset of the suspending force when the blade is twisted and thus an induced counteracting torsion.*
- *no exact perpendicular downwards facing force*
- *inclination of the blade*

*Regarding point 2, the offset of the load application point of the suspending cable from the numerically calculated shear center after twisting the blade, yield to 0.9%, 3.0% and 5.3% of the respective lever for the downwards facing force on LF2, LF3 and LF4, respectively. Theoretically, expecting a similar force pulling upwards as downwards, the induced torsion is reduced by the same relative values for the respective load cases MZLF2, MZLF3 and MZLF4.*

*The magnitude of the induced torsion was designed to be the maximum allowable torsion (respecting saftey margins) at the particular cross-section, rather then a possible bend-twist induced torsion magnitude. This was motivated by the certification idea of proofing the maximum torsion.*

*Nevertheless, overall the aforementioned errors do not practically affect the validation of the model, as all DWS and load cables are modeled as LINK11 elements. With all attachment points modeled at their correct global location of the test setup. This assures that the forces and displacement measurement direction is always correct throughout the test, all under the assumption that the model behaves the same as the real blade. Thus no corrections of any kind to measurements or FE results were applied."*

**Detailed comments:**

**Line 26-27: The authors state that 3D modelling is required to obtain a more reliable blade design. Please emphasise which structural behaviour cannot be captured by crosssection analysis tools – compelling the use of 3D FE models. For instance the inability of the stepwise prismatic approach to capture longitudinal geometric variations, particularly the effect of taper or any other local discontinuities, local buckling analysis, decoupled cross-sections etc.**

*"Nevertheless, at a final stage 3D FE analyses have to be performed in order to obtain a reliable blade design and account for structural details, e.g., adhesive joints, longitudinal geometric discontinuities, ply drops, or local buckling analysis, which are not considered in a 2D-FE-analysis."*

**Line 69: Typo / reference missing**

*Corrected.*

**Line 184: Please explain the meaning of a qualitatively satisfying mesh density. Especially in a validation process of local strains predicted by a FE model as presented in the paper, a mesh convergence study is crucial. How can the authors be confident that the deviations owing to local mesh discretization are indeed negligible?**

*"A mesh convergence study was performed in advance to ensure a satisfying mesh density. As stated before the purpose is to validate primarily the global blade behaviour and only secondary the local response. Therefore, no local mesh refinement will be performed, but the overall mesh density should yield acceptable convergence even at local level. Taking this into account, the convergence was first based on the global blade response in terms of total mass, center of gravity, tip deflection and the first 10 natural frequencies. Secondary the nodal strain results are examined for convergence at several position covering the whole blade. The element dimensions are halved each step. At the finally chosen mesh size the deviations to the next step are: global responses < 1.5%, strains < 2.1 μ-strains. It has to be stated, that for exact local strain measurements a modeling approach with solid-shells or layered solid elements is required to replicate the correct and detailed geometry of the structure."*

**Caption Fig.5: It is not entirely clear what an erroneous shear web adhesive joint indicates. Please explain more thoroughly.**

*The figure is extended by a marking in the picture highlighting the failure in the bonding. Additionally, the caption is extended by:*

*"The width should cover the complete web flange and the designed thickness is 9 +- 3 mm, however, the real thickness is measured to 33 mm."*

**Lines 424 – 435: The authors state at several places that they do not have a feasible explanation for the strain deviation. It would be important to get the authors opinion which strategy they suggest in order to shed light on this issue (maybe in the conclusions). The authors state that the vanishing sandwich core material caused a numerical strain peak. It would be interesting to know the authors opinion on how to practically deal with local material discontinuities in numerical modelling strategies.**

*See the comment to your last general comment.*

***Figures 12 and 13: Needs a drawing indicating the location of the SGs and their direction.***

*The cross-sectional sensor position is illustrated in two figure for each cross section in the appendix.*

*Dear Sarah Barber,*

*The authors gratefully thank you for taking the time and provide your very thorough and constructive review. We believe that the paper has been improved significantly by your suggestions. Please find our answers to all your comments, by either corrected or added text sections or comments on your suggestions. We hope to have adequately accommodated all your concerns.*

ABSTRACT

- Line 4: "However, there is a lack of test results for which the blade data are completely available": it's not entirely clear what this means. Please be more specific about which data you are talking about.

*"… lack of test results for which also the finite element model with blade geometry and layup as well as the test documentation and results are completely available."*

- Line 5: Please explain why this "one particular methodology" is validated. Is it the most common / important one?

*"… this paper is to validate the presented fully parameterized blade modeling methodology that is implemented in an in-house model generator, and to provide respective test results for validation purpose to the public. This methodology includes parameter definition based on splines for all design and material parameters, which enables fast and easy parameter analysis."*

- Line 6: Please explain why providing test results to the public for an in-house methodology is important / interesting.

*The test results are not limited to our methodology or tool validation, but shall be provided for the complete community. The thorough documentation of the test procedure and results should be interesting for anyone trying to validate a model generator. The necessary blade information can be made available upon request and will be published in a repository in near future.*

- Line 7: Why not mention the name of the code here?

*Thank you for your opinion. We prefer to keep the abstract as neutral as possible. Moreover, as the software does not play any significant role for this work, the name is omitted. However, it is named later in the work.*

- Line 8: Again, it's not clear to me exactly what "all relevant data" are. Same for the next sentence: "some data have been measured" doesn't really make sense to me.

*"A static full scale blade test is performed, which is used as the validation reference. All information, e.g., on sensor location, displacement and strains, are available to reproduce the tests."*

- Line 9: Why not mention "according to IEC 61400-23" here?

*The reference is included now.*

- Line 13: Please quantify "good".

*Concerning another later comment, we have defined validation tolerances. In the abstract we replace the good agreement by:*

*"Overall, the results meet the defined validation tolerances during bending…"*

- Line 14: Why and how do you conclude that "good" comparisons are equal to "validated"?

*With respect to the previous statement, we rate the modelling strategy as validated, excluding the torsional behaviour, which is not traced back to the modelling strategy, but the shell-element capability of predicting torsion.*

**1. INTRODUCTION**

- Generally: The "story" in this section is unclear. I suggest splitting this section into "Motivation", "State-of-the-art" and "Objectives of this paper", and move the paragraph starting at line 50 to the "Objectives of this paper" section, together with the paragraph starting at line 64.

*The introduction was split into:*

1. *State-of-the-art 3D Finite-Element Modeling of WindTurbine Blades*
2. *Objectives of this paper*
3. *Outline*

*And the paragraphs were sorted according to your suggestion.*

- Line 17: "designated": do you mean "expected"?

*The wording was corrected.*

- Line 20: I suggest writing "…one time only per blade design"

*The wording was corrected.*

- Lines 26-27: Does "have to be performed" mean that this has to be done because of the standard or in order to reach the required accuracy, in which case what is the required accuracy and why?

*Its not necessarily about accuracy, rather about not modelling or predicting certain phenomena in 2D.*

*"Nevertheless, at a final stage 3D FE analyses have to be performed in order to obtain a reliable blade design and account for structural details, e.g., adhesive joints, longitudinal geometric discontinuities, ply drops, or local buckling analysis, which are not considered in a 2D-FE-analysis."*

- Line 29: I don't really understand how automated model creation avoids model errors. Please elaborate on this.

*"... avoiding modeling errors caused by the user during a manual model creation."*

- Line 30: "neglecting details on structural information": such as?

*Expression replaced by:*

*"... QBlade for example focuses on the aerodynamic blade design, applying only an Euler-Bernoulli beam approach for the structure."*

- Line 32: what do you mean by a "proper composite definition"? Please explain this better

*Expression extended to:*

*"... taking into account a composite layup definition ..."*

- Line 36: specify what you mean by a "high discretization of stations" (how many?)

*Expression extended to:*

*"... high discretization of stations along the blade span (e.g., 45 stations for a 20~m blade) ..."*

- Line 41: I guess you mean here "the best compromise between accuracy and costs" rather than "a solution at minimum costs"??

*The wording was corrected.*

- Line 55: here you need to say why the method presented here is better, i.e. is it just more accurate or is it also more computationally efficient, thus offering a better compromise between costs and accuracy?

*This bullet list is added to the objective section of the Introduction:*

*"The presented method combines and extends several aspects of the different aforementioned software packages. The benefits are:*

- *it generates airfoils independent from any neighbouring geometry and uses the relative thickness distribution to position these along the span. This assures the geometry distribution, as this avoids any overshoot due to spanwise geometry interpolation.*
- *any parameter which may vary over the radius can be defined as spline, e.g., relative blade thickness, layer thickness, material density or stiffness.*
- *it enables flexible and easy parameter studies due to the simple parameter variation based on splines.*
- *it is designed for research, as different modules can be easily replaced by an alternative code, e.g., airfoil interpolation, adhesive modeling.*
- *it generates an FE-Model in MATLAB and provides already an interface to ANSYS APDL and BECAS, however interfaces to other FE-software can easily be implemented."*

- Line 56: This is important because..? Please elaborate.

*Included in the previously mentioned bullet list.*

- Lines 77-78: "Generally good agreement..." Please remove this sentence. It doesn't belong in the introduction, just the conclusions and the abstract!

*Sentence removed from introduction.*

2.MODEL CREATION FRAMEWORK

- Line 87: please specify what you mean by "efficient" (quick? easy?).

*"... enables ... design parameter variations in an easy way .."*

- Lines 100-110: it is not that clear which sentences correspond to which blocks in the figure. Please help readers by putting the relevant block name in brackets when it's not obvious, i.e. "First, the blade segmentation, i. e. the discretization in span-wise direction, is defined ("eval. CS geometry") (note -I don't even know if that's correct). Actually, it would be even better if you could number the blocks and refer

to them in the text.
- Lines 111-117: same comment as above.
- Lines 123-131: same as above.

*Thank you for that suggestion. The blocks in the flow charts were extended by indicies, which are used to refer them in the description.*

- Line 147: "there are no relative displacements between the master and the slave nodes". I'm not an expert on this: are there any consequences of this? Is it OK to do this? Maybe elaborate here.

*"The rigid connection implies that the deformation of the blade at the load frames is restricted. Similar to the real blade test according to IEC-64100, the load frames neighbouring blade sections cannot be evaluated, as the structural response is influenced by the quasi-rigid constraints."*

**3. MODELLING OF THE TEST BLADE**

- Line 165: please quantify "negligibly small" (and how you decide what "neglible" is in this context)

*"deviations in the outer geometry are negligibly small (Chord length < 10 mm and thickness < 8 mm)"*

- Line 173: put 5.2m in terms of relative position (xR) in brackets too. Also, indicate if the pressure side is the top or bottom (it's not obvious from that picture and we don't know which direction the blade is moving in :-)).

*The additions are added in parenthesis at the respective position and all other locations in the rest of the text.*

- Lines 173-175: be more specific about these deviations. Is this picture just an example of one deviation? Were all sections like this or just this one?

*"This was actually found throughout the whole blade, where the thickness varied between 20 and 30 mm. The design defined a thickness of 9 +- 3 mm."*

- Line 175: "is much thicker than specified in the design"...please explain how the picture shows this (is it the green part that is sticking out? and mark it on the picture!

*The bonding is marked in the picture and referred to in the text.*

**4. TEST DESCRIPTION AND VIRTUAL MODELLing**

- Line 198: The sentence " the weight of the blade bolts was subtracted from the

total mass." doens't really make sense: you don't subtract a weight from a mass. Be careful to be consistent with weight and mass (weight = mass x g).

*Thank you for highlighting this error, which was corrected!*

- Line 219: It's a bit strange that you mention "fatigue test" here as if it was obvious that you were carrying out "the fatigue test", whereas it is actually the first time you mention the word. "The fatigue test" therefore needs to be first introduced and described.

*The expression about the fatigue test was erased, as it is not a necessary information for this paper and raises confusion as you have stated.*

- Line 225: Explain why "almost horizontally" and what the consequences of this "almost" are.

*The "almost" was erased as it is described in the upcoming sections.*

- Line 238: Error in what? Displacement? Strain? And 0.5% of what?

*"Preliminary verifications showed that the corresponding error in the displacement is less than 0.5 % with respect to a simulation that accounts for gravitation throughout the whole simulation."*

- Line 264: again, please write what these radii are in terms of % along the span.

*Done.*

- Line 297: I'm not entirely sure what you mean by "the torsional moment is not parallel to the pitch axis" - that the moment is not applied around the pitch axis? Surely what you mean is that the applied forces are not perpendicular to the blade chord? Why does this influence the axis around which the moment is applied? Also, please be more specific about the descriptions here. Why did you assume that the location is 30 m above the ground (why not 10, or 100?) What is the consequence of this assumption?

*"Because the blade is still mounted at a block angle of 7.5° the torsional moment is not fully aligned with the pitch axis, as the forces do not act exactly in the cross sectional plane."*

*The correct location is at 18 m above ground. However, the results did not change, as the angle deviation of the rope suspending the blade is relatively small <1 deg, compared to the 30 m location.*

*"... location is 18m above the corresponding load point (y-direction, approximately crane height). By this, deflections parallel to the floor, due to load application, would only result in small angle deviation of the perpendicular force."*

- Line 303: "....as discussed in section 3"

- Section 4.4: This belongs in Section 3. I don't get why you have described some of the segment measurements in Section 3 and some here.

*This was added in Section 3 to clarify why the cross sectional discussion is split into different sections.*

*"During the manufacturing procedure and the latter testing, several different properties of the blade are captured. From these, the FE model only covers the geometrical deviations such as the chord, thickness and adhesive geometry deviations. However, mass and stiffness adaption, to meet the measured natural frequencies and masses, or displacements will not be covered, as this would demand a thorough model updating procedure, which would go beyond the scope of this work. Therefore, this Section will refer to the geometrical measurements and the rest will be covered in Section 4."*

5. COMPARISON OF RESULTS

- Line 325: Please explain why you didn't first obtain the mass in the numerical model and then add mass until these values matched, before continuing with the simulations.

*See previous comment.*

- Line 325: Explain how you estimated the measurement uncertainties.

*"All measurement uncertainties are based on given sensor uncertainties and taking the worst case scenario in the combination of those."*

- Line 328: Quantify "perfectly" (i.e. to the nearest two decimal places).

"The location of the CoG matches perfectly in the chord direction, i.e., with a precision to the nearest two decimal places."

- Line 332: How do you decide what "acceptable" is?

*Thank you for highlighting this significant weakness of the paper. However, it is difficult to find a reference for validation tolerances, as these are somehow subjective/arbitrary. We have tried to define some reasonable thresholds based on a presentation and respecting manufacturing tolerances.*

*"In order to rate the validity of the model, it is necessary to identify specific thresholds. However, these are hardly found in literature, especially as different applications and fidelity levels may demand other thresholds. Exemplary, Safarian.2015reports validation requirements for finite element analysis according to the Federal Aviation Regulation of the U.S. government, where a displacement deviation of <5% between the simulation and experiment is typically acceptable for global effects and local effects measured in form of strains allow for a maximum of 10% deviation. Whereas strains exceeding these values require a re-evaluation of the model. These regulations refer to aviation applications, which also apply finite element shell models for the analysis comparable to our use case. Therefore, we will apply a 5% threshold for global displacements, whereas a 10% threshold will be applied on the cross-sectional strain results. These margins should also cover measurement uncertainties, as the DWS and the strain gauges offer a quite narrow uncertainty band, 0.6% and $2\%$, respectively. Thresholds for masses are harder to define as these depend on the measurement setup, in our case with up to 2.5% uncertainty. Plus, not all additional masses were correctly documented and thus not modeled. Same problem holds for natural frequencies ω, following $\omega = \sqrt{\frac{k}{m}}$ and respecting unknown mass variation and typically a maximum of 5% material tolerances (including density and stiffness according to private communication with manufacturers), it is also hard to define thresholds for the frequencies. Therefore, both mass and frequencies will be discussed individually."*

**- Lines 332-345: Can you give the reader some idea of how good these comparisons are? Do other comparisons of experiments and simulations show similar results? Or are yours much better or worse than others?**

*I have found some comparable modal simulation of the same blade:*

*"Similar deviation ranges can be found in Knebusch.2020 for the same blade but with a different model, with errors between 1.8% to 9.7% for flapwise and edgewise modes and up to 22% for the torsion mode."*

**- Line 378: Clarify exactly what you mean with "due to the wrong twist". Wrongly measured? Wrongly simulated? Wrongly assumed?**

*The wording twist is somehow confusable with the structural designed twist of the blade. Therefore it is replaced by "rotation" or "rotation around the z-axis". Additionally the expression "due to the wrong twist" is corrected to:*

*"... due to the wrong predicted rotation alogn the z-axis."*

- Line 410: "Such high errors during torsional loading may base on the shell element with a node offset to the exterior surface used for this model": Do you have any suggestions to improve / solves this?

*This is included in the conclusion:*

*"However, modern flexible blade design, which is driven to it's material and structural integrity limits and includes intentional torsion for load alleviation, requires accurate predictions for all load cases in order to be reliable. Looking a step further, especially fatigue damage calculation, therefore, needs correct strain or stress predictions of the models. The authors currently work on evaluating blade modeling by means of solid elements and/or solid shell elements. Although, we loose computational efficiency of the shell element models, this way, the accuracy in torsional response should be improved significantly, as well as the correct representation of geometrical shape and 3D tapering can be realized. This should shed light on the discrepancy in torsion"*

- Line 425: "because the sandwich core material vanishes towards the trailing edge": Can you explain what this means please? Does this happen in the simulations but not in reality?

*"... the sandwich core material vanishes suddenly towards the trailing edge, due to the shell elements and their missing capability of tapering single materials in their sections as done in the real layup."*

- Line 441: Explain what you mean by "a wrong calibration or malfunction of the strain gauge"

*"Unlike the edge-wise case a failure of the strain gauge at S = 0.3 was recorded in the experiment, which can be seen in the discontinuity of the experimental results."*

- Section 5.5: Again, it's not clear to me why these couldn't be "calibrated" before running/repeating the simulations

*Commented before*

Technical corrections

*Thank you for your technical corrections, which were all considered in the revision. Additionally, the paper has gone through a proof-reading process.*